# Mapping spatial patterns to energetic benefits in groups of flow-coupled swimmers

Sina Heydari[1†], Haotian Hang[1†], Eva Kanso[1,2]*

[1]Department of Aerospace and Mechanical Engineering, University of Southern California, Los Angeles, United States; [2]Department of Physics and Astronomy, University of Southern California, Los Angeles, United States

## eLife assessment

This **fundamental** study provides a modeling regime that provides new insight into the energy-preservation parameters among schooling fish. The strength of the evidence supporting observations such as distilled dynamics between leading and lagging schooling fish which are derived from emergent properties is **compelling**. Overall, the study provides exciting insights into energetic coupling with respect to group swimming dynamics.

*For correspondence:
kanso@usc.edu

†These authors contributed equally to this work

Competing interest: The authors declare that no competing interests exist.

**Abstract** The coordinated motion of animal groups through fluids is thought to reduce the cost of locomotion to individuals in the group. However, the connection between the spatial patterns observed in collectively moving animals and the energetic benefits at each position within the group remains unclear. To address this knowledge gap, we study the spontaneous emergence of cohesive formations in groups of fish, modeled as flapping foils, all heading in the same direction. We show in pairwise formations and with increasing group size that (1) in side-by-side arrangements, the reciprocal nature of flow coupling results in an equal distribution of energy requirements among all members, with reduction in cost of locomotion for swimmers flapping inphase but an increase in cost for swimmers flapping antiphase, and (2) in inline arrangements, flow coupling is non-reciprocal for all flapping phase, with energetic savings in favor of trailing swimmers, but only up to a finite number of swimmers, beyond which school cohesion and energetic benefits are lost at once. We explain these findings mechanistically and we provide efficient diagnostic tools for identifying locations in the wake of single and multiple swimmers that offer opportunities for hydrodynamic benefits to aspiring followers. Our results imply a connection between the resources generated by flow physics and social traits that influence greedy and cooperative group behavior.

## Introduction

Flow interactions are thought to allow flying and swimming animals to derive energetic benefits when moving in groups (*Weihs, 1973*). However, direct assessment of such benefits is challenging, chiefly because animal groups do not generally conform to regular patterns – individuals in these groups dynamically change their position relative to their neighbors (*Partridge and Pitcher, 1979*; *Svendsen et al., 2003*; *Marras et al., 2015*; *Ashraf et al., 2017*; *Mirzaeinia et al., 2020*). Also, because direct energetic measurements in moving animals, flying or swimming, are notoriously difficult and often unreliable as proxy for hydrodynamic energy savings (*Herskin and Steffensen, 1998*; *Killen et al., 2012*; *Marras et al., 2015*; *Li et al., 2020*; *Zhang and Lauder, 2023a*; *Thandiackal and Lauder, 2023*; *Zhang and Lauder, 2023b*; *Marras and Porfiri, 2012*). These difficulties hinder a direct mapping from

the spatial pattern of the group to the energetic benefits or costs incurred at each position within the group.

An understanding of how the spatial arrangement of individuals within a group influences their cost of locomotion can provide insights into the evolution of social structures, resource allocation, and overall fitness of each individual in cooperative activities such as foraging, mating, and evasion (*Whitehead, 1997*; *Couzin et al., 2002*; *Couzin and Krause, 2003*; *Croft et al., 2008*; *Bajec and Heppner, 2009*; *Farine and Whitehead, 2015*; *Jiao et al., 2023*). It could also guide the design of bio-inspired engineering systems and algorithms that steer groups of entities, such as swarms of autonomous robotic vehicles, underwater or in flight, that collaborate to achieve a desired task while minimizing energy consumption and improving the overall system efficiency (*Leonard and Fiorelli, 2001*; *Zhu et al., 2019*; *Coppola et al., 2020*; *Li et al., 2021*; *Berlinger et al., 2021*).

To understand the potential energetic benefits of group movement, various direct and indirect approaches have been employed. *Li et al., 2020*, associated energy savings in pairs of flapping robotic swimmers with a linear relationship between their flapping phase lag and relative distance. Based on this, a strategy, called *vortex phase matching*, was extrapolated for how fish should behave to maximize hydrodynamic benefits: a follower fish should match its tailbeat phase with the local vorticity created by a leader fish. Pairs of freely swimming fish seemed to follow this linear phase-distance relationship even with impaired vision and lateral line sensing, i.e., in the absence of sensory cues about their relative position and neighbor-generated flows. Interestingly, the same linear phase-distance relationship was uncovered independently in flapping hydrofoils and accredited solely to flow interactions (*Zhu et al., 2014*; *Newbolt et al., 2019*; *Newbolt et al., 2022*). It is therefore unclear whether vortex phase matching is an active strategy, mediated by sensing and feedback control, that fish employ to minimize energy expenditure, or if it arises passively through flow interactions between flapping swimmers. Importantly, active or passive, it is unclear if this strategy scales to groups larger than two.

In an effort to directly gauge the energetic benefits of schooling, metabolic energetic measurements were recently performed in solitary and groups of eight fish, and impressive energetic savings were attributed to schooling compared to solitary swimming when the fish were challenged to swim at high speeds (*Zhang and Lauder, 2023b*). Lamentably, the study made no mention of the spatial patterns assumed by these physically thwarted individuals (*Zhang and Lauder, 2023b*). In an independent previous study (*Ashraf et al., 2017*), changes in spatial patterns and tailbeat frequencies were reported in similar experiments, albeit with no energetic measurements. Specifically, *Ashraf et al., 2017*, showed that, when challenged to sustain higher swimming speeds, the fish in a group rearranged themselves in a side-by-side pattern as the speed increased, presumably to save energy.

Taking together the results of *Zhang and Lauder, 2023b*; *Ashraf et al., 2017*, are we to conclude that side-by-side formations are more energetically beneficial than, say, inline or diagonal formations? The answer is not simple! The metabolic measurements of *Marras et al., 2015*, in a school of eight fish report that side-by-side formations, though beneficial, produce the least energetic savings compared to diagonal formations (*Weihs, 1973*). In an experimental study of a single fish interacting with a robotic flapping foil, the freely swimming fish positioned itself in an inline formation in the wake of the flapping foil, supporting the hypothesis that swimming in the wake of one another is an advantageous strategy to save energy in a school (*Thandiackal and Lauder, 2023*). Why did the fish in the experiments of *Ashraf et al., 2017*, self-organize in a side-by-side formation when challenged to swim at higher swimming speeds?

The answer is not simple because ample hydrodynamic mechanisms for energy savings in fish schools have been stipulated for each possible configuration – side-by-side, inline, and diagonal (see, e.g., Figure 1 of *Zhang and Lauder, 2023b*) – but no assessment is provided of the relative advantage of these configurations. For example, side-by-side formations, where fish mirror each other by flapping antiphase, are thought to create a wall-like effect that reduces swimming cost (*Zhang and Lauder, 2023b*; *Fish and Hui, 1991*). A fish swimming in the wake between two leading fish encounters a reduced oncoming velocity, leading to reduced drag and thrust production (*Weihs, 1973*). Inline formations, where fish swim in tandem, are thought to provide benefits to both leader and follower, by an added mass push from follower to leader (*Fish and Hui, 1991*; *Usherwood et al., 2011*) and a reduced pressure on the follower (*Kurt and Moored, 2018*). All of these mechanisms can in principle be exploited by schooling fish as they dynamically change their relative spacing in

the group. But are these mechanisms equally advantageous? Or is there a hierarchy of hydrodynamic benefits depending on the relative position within the school? The literature offers no comparative analysis of the energetic savings afforded by each of these configurations.

The study of *Marras et al., 2015*, is arguably the closest to addressing this question, but, to map the energetic benefits for pairwise configurations, the authors employed statistical averages in a school of eight fish, thus inevitably combining the various hydrodynamic mechanisms at play and cross-polluting the estimated benefits of each configuration. A cleaner analysis in pairs of flapping foils shows that these relative positions – side-by-side, inline, and diagonal – all emerge spontaneously and stably due to flow interactions (*Newbolt et al., 2022*), but provides no method for estimating the energetic requirements of these formations, let alone comparing them energetically. Even vortex phase matching makes no distinction between side-by-side, inline, or diagonal pairs of fish (*Li et al., 2020*). It simply postulates that an unknown amount of energetic benefit is acquired when the linear phase-distance relationship is satisfied. Thus, to date, despite the widespread notion that group movement saves energy, a direct comparison of the energetic savings afforded by different spatial formations remains lacking. Importantly, it is unknown whether and how the postulated benefits scale with increasing group size.

Here, to circumvent the challenges of addressing these questions in biological systems, we formulate computational models that capture the salient hydrodynamic features of single and pairs of swimming fish. Namely, we represent each fish as a freely swimming hydrofoil undergoing pitching oscillations about its leading edge. A single flapping hydrofoil shares many hydrodynamic aspects with its biological counterpart, including an alternating, long-lived pattern of vorticity in its wake (*Triantafyllou et al., 1993*; *Triantafyllou et al., 2000*; *Lauder et al., 2011*; *Verma et al., 2017*; *Smits, 2019*). These similarities have been demonstrated repeatedly, within biologically relevant ranges of flapping parameters (*Triantafyllou et al., 1993*; *Taylor et al., 2003*), for different geometries (*Von Ellenrieder et al., 2003*; *Taira and Colonius, 2009*; *Green et al., 2011*; *Ayancik et al., 2019*), material properties (*Moored et al., 2014*; *Quinn et al., 2014*; *Ayancik et al., 2019*), and flapping kinematics (*Tytell and Lauder, 2004*; *Kern and Koumoutsakos, 2006*; *Hultmark et al., 2007*). In this study, we show, based on our own simulations and by conducting a thorough literature survey, that flow interactions, with no sensing and feedback control, lead to emergent formations that preserve the linear phase-distance relationship uncovered independently in live and robotic fish (*Li et al., 2020*; *Thandiackal and Lauder, 2023*) and in flapping hydrofoils (*Zhu et al., 2014*; *Newbolt et al., 2019*; *Newbolt et al., 2022*). This relationship is preserved irrespective of geometry (*Ramananarivo et al., 2016*; *Newbolt et al., 2019*; *Newbolt et al., 2022*), material properties (*Kim et al., 2010*; *Zhu et al., 2014*; *Peng et al., 2018*; *Arranz et al., 2022*), and flapping kinematics (*Heydari and Kanso, 2021*; *Kurt et al., 2021*). The universality of this relationship serves as strong validation of our models and anchors our subsequent exploration of the opportunities for hydrodynamic benefits available in a given flow field.

Importantly, we go beyond two swimmers to investigate flow interactions in larger groups and find that inline formations *differentially* distribute hydrodynamic savings to members within the school, favoring trailing swimmers, but only up to a certain school size, while side-by-side formations *equally* distribute hydrodynamic savings and scale to arbitrary number of swimmers without loss of cohesion. Our findings provide a direct mapping from the school's spatial pattern to the energetic savings experienced by its members. Importantly, our results raise an interesting hypothesis that the dynamic repositioning of members within a fish school could be driven by greed and competition to occupy hydrodynamically advantageous positions and open up opportunities for analyzing the role of flow physics in the evolution of cooperative versus greedy behavior in animal groups.

## Results

### Mathematical models of flow-coupled flapping swimmers

Inspired by the experiments of *Newbolt et al., 2019*; *Li et al., 2020*, we study self-organization in the context of flapping swimmers, coupled passively via the fluid medium, with no mechanisms for visual (*McKee et al., 2020*; *Guthrie, 1986*; *Fernald, 1989*; *Douglas and Djamgoz, 2012*; *Lombana and Porfiri, 2022*), flow sensing (*Engelmann et al., 2000*; *Ristroph et al., 2015*; *Colvert and Kanso, 2016*; *Hang et al., 2023*), or feedback control (*Verma et al., 2018*; *Jiao et al., 2021*; *Li et al., 2021*;

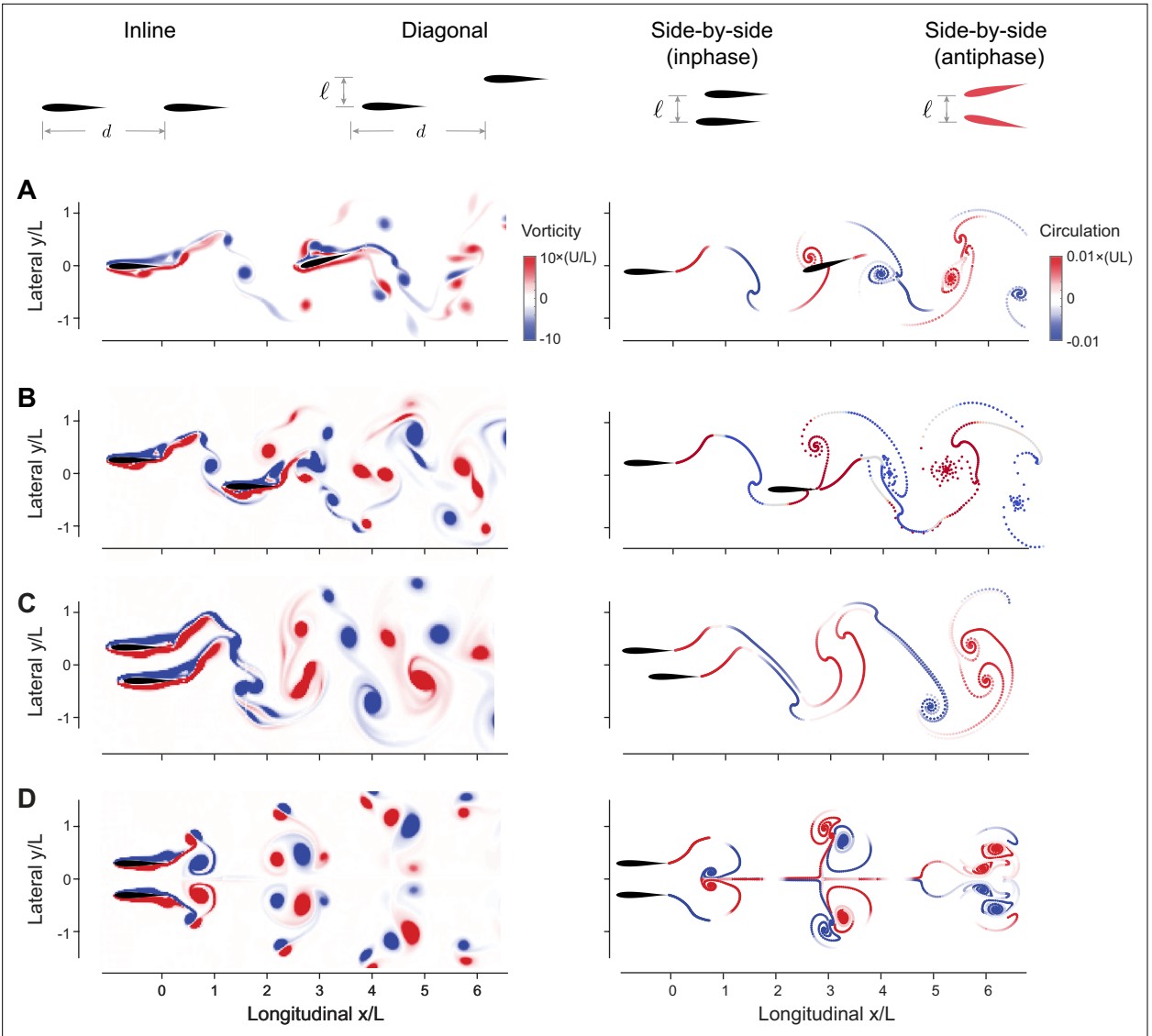

**Figure 1.** Flow-coupled swimmers self-organize into stable pairwise formations. (**A**) Inline ($\ell = 0, \phi = \pi/2$), (**B**) diagonal ($\ell = L/2, \phi = 0$), (**C**) inphase side-by-side ($\ell = L/2, \phi = 0$), and (**D**) antiphase side-by-side ($\ell = L/2, \phi = \pi$) in computational fluid dynamics (CFD) (left) and vortex sheet (VS) (right) simulations. Power savings at steady state relative to respective solitary swimmers are reported in *Figure 3*. Parameter values are $A = 15°$, Re = $2\pi\rho A f L/\mu = 1645$ in CFD, and $f\tau_{\mathrm{diss}} = 2.45$ in VS simulations. Corresponding hydrodynamic moments are given in *Figure 1—figure supplement 4*. Simulations at different Reynolds numbers and dissipation times are given in *Figure 1—figure supplements 1–3*.

The online version of this article includes the following figure supplement(s) for figure 1:

**Figure supplement 1.** Computational fluid dynamics (CFD) simulations of pairwise formations.

**Figure supplement 2.** Vortex sheet (VS) simulations of pairwise formations.

**Figure supplement 3.** Computational fluid dynamics (CFD) and vortex sheet (VS) simulations at various flow properties.

**Figure supplement 4.** Hydrodynamic moment acting on pairs of flapping swimmers.

*Figure 1*). The swimmers are rigid, of finite body length $L$ and mass per unit depth $m$, and undergo pitching oscillations of identical amplitude $A$ and frequency $f$ in the $(x, y)$-plane of motion, such that the pitching angle for swimmer $j$ is given by $\theta_j = A \sin(2\pi f t + \phi_j)$, $j = 1, 2, \ldots, N$, where $N$ is the total number of swimmers. In pairwise interaction, we set $\phi_1 = 0$ and $\phi_2 = -\phi$, with $\phi$ being the phase lag between the oscillating pair. We fixed the lateral distance $\ell$ between the swimmers to lie in the range $\ell \in [-L, L]$, and allowed the swimmers to move freely in the $x$-direction in an unbounded two-dimensional fluid domain of density $\rho$ and viscosity $\mu$.

When unconstrained, the swimmers may drift laterally relative to each other, as illustrated in dipole models (*Tsang and Kanso, 2013*; *Kanso and Cheng Hou Tsang, 2014*) and high-fidelity simulations of undulating swimmers (*Gazzola et al., 2014*; *Verma et al., 2018*). However, this drift occurs at a slower time scale than the swimming motion, and can, in principle, be corrected by separate feedback control mechanisms (*Zhu et al., 2022*). Here, we focus on the dynamics in the swimming direction.

Hereafter, all parameters are scaled using the body length $L$ as the characteristic length scale, flapping period $T = 1/f$ as the characteristic time scale, and $\rho L^2$ as the characteristic mass per unit depth. Accordingly, velocities are scaled by $Lf$, forces by $\rho f^2 L^3$, moments by $\rho f^2 L^4$, and power by $\rho f^3 L^4$.

The equations governing the free motion $x_j(t)$ of swimmer $j$ are given by Newton's second law (here, the downstream direction is positive),

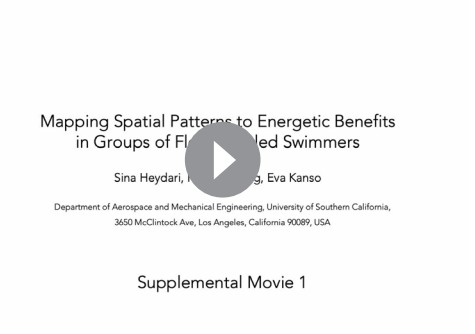

Video 1. Pairs of flow-coupled swimmers self-organize into stable inline, staggered, or side-by-side formation. Energetic benefits depend on the spatial pattern. Results based on flapping foils in computational fluid dynamics (CFD) simulations.
https://elifesciences.org/articles/96129/figures#video1

$$m\ddot{x}_j = -F_j \sin\theta_j + D_j \cos\theta_j. \tag{1}$$

The hydrodynamic forces on swimmer $j$ are decomposed into a pressure force $F_j$ acting in the direction normal to the swimmer and a viscous drag force $D_j$ acting tangentially to the swimmer. These forces depend on the fluid motion, which, in turn, depends on the time history of the states of the swimmers.

To maintain their pitching motions, swimmers exert an active moment $M_a$ about the leading edge, whose value is obtained from the balance of angular momentum. The hydrodynamic power $P$ expended by a flapping swimmer is given by $P = M_a \dot{\theta}$.

To compute the hydrodynamic forces and swimmers' motion, we used two fluid models (*Figure 1—figure supplements 1–3*). First, we employed a computational fluid dynamics (CFD) solver of the Navier-Stokes equations tailored to resolving fluid-structure interactions (FSI) based on an adaptive mesh implementation of the immersed boundary method (IBM) (*IBAMR, 2024*; *Griffith et al., 2007*; *Bhalla et al., 2013*). Then, we solved the same FSI problem, in the limit of thin swimmers, using the more computationally efficient inviscid vortex sheet (VS) model (*Nitsche and Krasny, 1994*; *Huang et al., 2016*; *Huang et al., 2018*; *Heydari and Kanso, 2021*; *Hang et al., 2022*). To emulate the effect

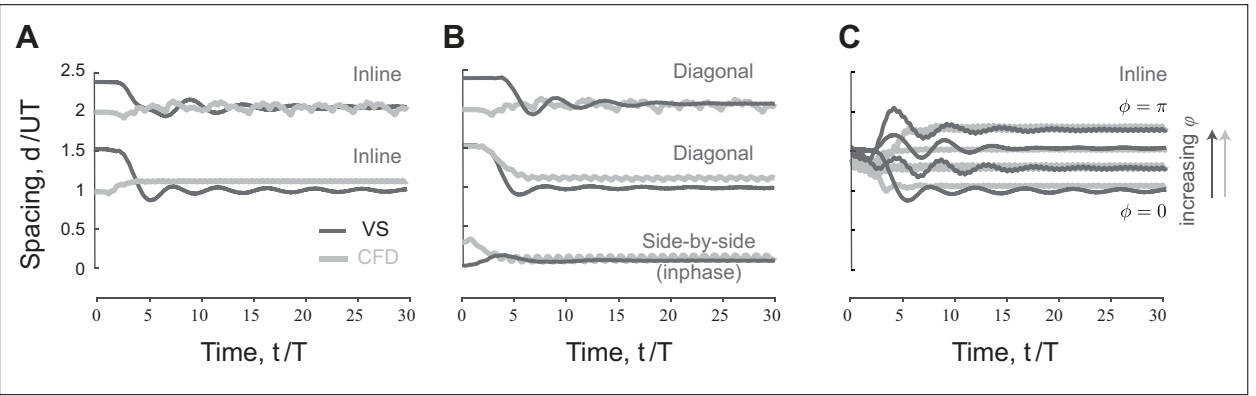

**Figure 2.** Emergent equilibria in pairwise formations. (**A**) Time evolution of scaled streamwise separation distance $d/UT$ for a pair of inline swimmers at $\phi = 0$. Depending on initial conditions, the swimmers converge to one of two equilibria at distinct separation distance. (**B**) At $\ell = L/2$, $d/UT$ changes slightly compared to inline swimming in (A). Importantly, a new side-by-side inphase equilibrium is now possible where the swimmers flap together at a slight shift in the streamwise direction. (**C**) Starting from the first equilibrium in (A), $d/UT$ increases linearly as we increase the phase lag $\phi$ between the swimmers.

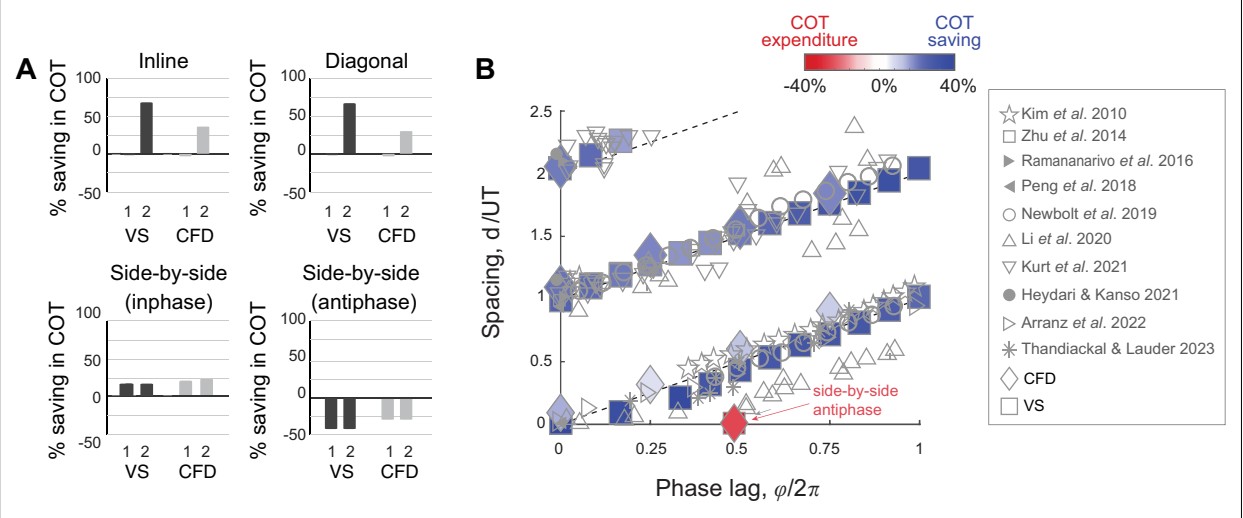

**Figure 3.** Hydrodynamic benefits and linear phase-distance relationship in pairwise formations. (**A**) Change in cost of transport compared to solitary swimmers for the inline, diagonal, side-by-side inphase, and side-by-side antiphase formations shown in *Figure 1*. (**B**) Emergent formations in pairs of swimmers in computational fluid dynamics (CFD) and vortex sheet (VS) models satisfy a linear phase-distance relationship, consistent with experimental (*Ramananarivo et al., 2016*; *Newbolt et al., 2019*; *Li et al., 2020*; *Kurt et al., 2021*; *Thandiackal and Lauder, 2023*) and numerical (*Kim et al., 2010*; *Peng et al., 2018*; *Heydari and Kanso, 2021*; *Arranz et al., 2022*) studies. With the exception of the antiphase side-by-side formation, swimmers in these formations have a reduced average cost of transport compared to solitary swimming.

of viscosity in the VS model, we allowed shed vorticity to decay after a dissipation time $\tau_{\mathrm{diss}}$; larger $\tau_{\mathrm{diss}}$ correlates with larger Reynolds number Re in the Navier-Stokes model; see SI for a brief overview of the numerical implementation and validation of both methods.

## Flow coupling leads to stable emergent formations

We found, in both CFD and VS models, that pairs of swimmers self-organize into relative equilibria at a streamwise separation distance $d$ that is constant on average, and swim together as a single formation at an average free-swimming speed $U$ (*Figures 1 and 2*). We distinguished four types of relative equilibria: inline, diagonal, side-by-side inphase, and side-by-side antiphase (*Figure 1*, *Video 1*).

Inline formations at $\ell = 0$ arise when the follower positions itself, depending on its initial distance from the leader, at one of many inline equilibria, each with its own basin of attraction (*Figure 2A*). These inline equilibria occur at average spacing $d$ that is approximately an integer multiple of $UT$, consistent with previous experimental (*Becker et al., 2015*; *Ramananarivo et al., 2016*; *Newbolt et al., 2019*) and numerical (*Zhu et al., 2014*; *Park and Sung, 2018*; *Peng et al., 2018*; *Dai et al., 2018*; *Heydari and Kanso, 2021*; *Arranz et al., 2022*) findings.

When offsetting the swimmers laterally at $\ell \neq 0$ (*Figure 2B*), the leader-follower equilibria that arise at $\ell = 0$ shift slightly but persist, giving rise to diagonal leader-follower equilibria (*Newbolt et al., 2022*). Importantly, at a lateral offset $\ell$, inphase swimmers ($\phi = 0$) that are initially placed side-by-side reach a relative equilibrium where they travel together at a close, but non-zero, average spacing $d \leq L$. That is, a perfect side-by-side configuration of inphase flapping swimmers is unstable but the more commonly observed configuration (*Li et al., 2020*) where the two swimmers are slightly shifted relative to each other is stable. This configuration is fundamentally distinct in terms of cost of transport from the mirror-symmetric side-by-side configuration that arises when flapping antiphase at $\phi = \pi$ (*Figure 3A*). Both side-by-side equilibria were observed experimentally in heaving hydrofoils (*Newbolt et al., 2022*), albeit with no assessment of the associated hydrodynamic power and cost of transport.

We next examined the effect of varying the phase $\phi$ on the emergent traveling formations. Starting from initial conditions so as to settle on the first equilibrium $d/UT \approx 1$ when $\phi = 0$, and increasing $\phi$, we found, in both CFD and VS simulations, that the spacing $d/UT$ at equilibrium increased with increasing $\phi$ (*Figure 2C*). This increase is linear, as evident when plotting $d/UT$ as a function of $\phi$ (*Figure 3B*). Indeed, in *Figure 3B*, we plotted the emergent average separation distance $d/UT$ as a function of

$\phi$ for various values of $\ell$. Except for the antiphase side-by-side formation, the linear phase-distance relationship $\phi/2\pi \propto d/UT$ persisted for $\ell \neq 0$.

The key observation, that pairs of flapping swimmers passively self-organize into equilibrium formations, is independent of both scale and fluid model. In our CFD simulations (*Figure 1—figure supplements 1 and 3*), we tested a range of Reynolds number $\mathrm{Re} = \rho UL/\mu$ from 200 to 2000, which covers the entire range of existing CFD simulations (*Becker et al., 2015*; *Arranz et al., 2022*), where $\mathrm{Re} \sim O(10^2)$, and experiments (*Becker et al., 2015*; *Newbolt et al., 2019*; *Newbolt et al., 2022*), where $\mathrm{Re} \sim O(10^3)$. In our VS simulations, we varied $\tau_{\mathrm{diss}}$ from $2.45T$ to $\infty$ (*Figure 1—figure supplements 2 and 3*). Note that the separation distance $d$ is scale-specific and increases with Re; at low Re, a compact inline formation is reached where the two swimmers 'tailgate' each other, as observed in *Peng et al., 2018*. However, the scaled separation distance $d/UT$ remains nearly constant for all Re and $\tau_{\mathrm{diss}}$ (*Figure 1—figure supplement 3*).

The fact that these equilibria emerge in time-forward simulations is indicative of stability (*Strogatz, 1994*). A more quantitative measure of linear stability can be obtained numerically by perturbing each equilibrium, either by applying a small impulsive or step force after steady state is reached (*Newbolt et al., 2022*) or by directly applying a small perturbation to the relative equilibrium distance between the two swimmers and examining the time evolution of $d$ and $F$ to quantify variations in hydrodynamic force $\delta F$ as a function of signed variations in distance $\delta d$ from the equilibrium (*Heydari and Kanso, 2021*). In either case, we found that the force-displacement response to small perturbations at each equilibrium exhibited the basic features of a linear spring-mass system, where $\delta F/\delta d$ is negative, indicating that the hydrodynamic force acts locally as a restoring spring force that causes the initial perturbation to decay and that stabilizes the two swimmers together at their equilibrium relative position. Larger values of $|\delta F/\delta d|$ imply faster linear convergence to the stable equilibrium and thus stronger cohesion of the pairwise formation. Results of this quantitative stability analysis are discussed in subsequent sections.

## Emergent formations save energy compared to solitary swimming

We evaluated the hydrodynamic advantages associated with these emergent formations by computing the hydrodynamic power $P_{\mathrm{single}}$ of a solitary swimmer and $P_j$ of swimmer $j$ in a formation of $N$ swimmers.

We calculated the cost of transport $\mathrm{COT}_j = P_j/mU$, of swimmer $j$ and the change in COT compared to solitary swimming $\Delta\mathrm{COT}_j = (\mathrm{COT}_{\mathrm{single}} - \mathrm{COT}_j)/\mathrm{COT}_{\mathrm{single}}$ (*Figure 3A*). We also calculated the average change in cost of transport $\Delta\mathrm{COT} = \sum_j^N \Delta\mathrm{COT}_j/N$ for each formation (*Figure 3B*). In all cases, except for the antiphase side-by-side formation, in both CFD and VS simulations, the swimmers traveling in equilibrium formations save power and cost of transport compared to solitary swimming. The savings are larger at tighter lateral spacing $\ell$.

For inline and diagonal formations, these hydrodynamic benefits are granted entirely to the follower, whose hydrodynamic savings can be as high as 60% compared to solitary swimming (*Figure 3A*; *Heydari and Kanso, 2021*). Intuitively, because in 2D flows, vortex-induced forces decay with the inverse of the square of the distance from the vortex location, flow coupling between the two inline or diagonal swimmers is non-reciprocal; the follower positioned in or close to the leader's wake interacts more strongly with that wake than the leader interaction with the follower's wake (*Figure 1—figure supplements 1A, B and 4A, B*).

In side-by-side formations, by symmetry, flow coupling between the two swimmers is reciprocal, or nearly reciprocal in inphase flapping (*Figure 1—figure supplements 1C, D and 4C, D*). Thus, hydrodynamics benefits or costs are expected to be distributed equally between the two swimmers. Indeed, for inphase flapping, the hydrodynamic benefits are shared equally between both swimmers. For antiphase flapping the cost is also shared equally (*Figure 3A*).

The biased distribution of benefits in favor of the follower in inline and diagonal formations could be a contributing factor to the dynamic nature of fish schools (*Svendsen et al., 2003*; *Mirzaeinia et al., 2020*). The egalitarian distribution of benefits in the inphase side-by-side formation could explain the abundance of this pairwise configuration in natural fish populations (*Li et al., 2020*) and why groups of fish favor this configuration when challenged to swim at higher speeds (*Ashraf et al., 2017*; *Lombana and Porfiri, 2022*).

**Table 1.** Data collection from published literature.

We calculated the Reynolds numbers based on swimming speed ($\text{Re}_U = \rho U L / \mu$) and flapping velocity ($\text{Re}_A = 2\pi \rho A f L / \mu$), where $f = 1/T$ is the flapping frequency. Missing values are either not applicable or not available.

| Reference | Study type | System | Distance | Phase | $\text{Re}_U$ | $\text{Re}_A$ |
|---|---|---|---|---|---|---|
| *Kim et al., 2010* | Numerical | Flapping flags | Head-to-head | $\phi \in [0, 2\pi]$ | 200 –400 | – |
| *Zhu et al., 2014* | Numerical | Heaving flexible foils | – | $\phi \in [0, \pi]$ | 500 | 200 |
| *Ramananarivo et al., 2016* | Experimental | Heaving foils | Tail-to-head | $\phi = 0$ | $10^3$ –$10^4$ | $10^2$ –$10^3$ |
| *Peng et al., 2018* | Numerical | Heaving flexible foils | Tail-to-head | $\phi = 0$ | 509 | 200 |
| *Newbolt et al., 2019* | Experimental | Heaving foils | Tail-to-head | $\phi \in [0, 2\pi]$ | $10^3$ –$10^4$ | $10^2$ –$10^3$ |
| *Li et al., 2020* | Experimental | Goldfish | Head-to-head | $\phi \in [0, 2\pi]$ | $10^5$ | $10^4$ |
| *Kurt et al., 2021* | Experimental | Pitching foils | Tail-to-head | $\phi \in [0, 2\pi]$ | 9950 | 18,850 |
| *Heydari and Kanso, 2021* | Numerical | Heaving plates | Tail-to-head | $\phi = 0$ | – | – |
| | | Pitching plates | Head-to-head | $\phi = 0$ | – | – |
| *Arranz et al., 2022* | Numerical | Heaving flexible foils (3D) | Tail-to-head | $\phi \in [0, 2\pi]$ | 176 | 200 |
| *Thandiackal and Lauder, 2023* | Experimental | Fish-foil interactions | Tail-to-head | $\phi \in [0, 2\pi]$ | $20 \cdot 10^3$ (foil) | $50 \cdot 10^3$ (foil) |
| | | | | | $40 \cdot 10^3$ (fish) | |
| *Becker et al., 2015\** | Experimental | Heaving foils | Tail-to-head | $\phi = 0$ ,$\pi$ | – | $10^2$ –$10^4$ |
| *Park and Sung, 2018\** | Numerical | Heaving flexible foils | – | $\phi \in [0, 2\pi]$ | 60 –1100 | 100 –1200 |
| *Dai et al., 2018\** | Numerical | Pitching + heaving foils | – | $\phi = 0$ ,$\pi$ | 440 | 600 |

\*Data not included in **Figure 3**.

## Linear phase-distance relationship in emergent formations is universal

To probe the universality of the linear phase-distance relationship, we compiled, in addition to our CFD and VS results, a set of experimental (*Ramananarivo et al., 2016*; *Newbolt et al., 2019*; *Li et al., 2020*) and numerical (*Kim et al., 2010*; *Peng et al., 2018*; *Heydari and Kanso, 2021*; *Kurt et al., 2021*; *Arranz et al., 2022*) data from the literature (*Table 1*). Data including CFD simulations of deformable flapping flags (☆) (*Kim et al., 2010*), (❏) (*Zhu et al., 2014*) and flexible airfoil with low aspect ratio (▷) (*Arranz et al., 2022*), physical experiments with heaving (◯) (*Ramananarivo et al., 2016*; *Newbolt et al., 2019*) and pitching (▽) (*Kurt et al., 2021*) rigid hydrofoils, fish-foil interactions (∗) (*Thandiackal and Lauder, 2023*), and fish-fish interactions (△) measured in pairs of both intact and visually and/or lateral line-impaired live fish (*Li et al., 2020*) are superimposed on *Figure 3B*. All data collapsed onto the linear phase-distance relationship $\phi/2\pi \propto d/UT$, with the largest variability exhibited by live fish with close streamwise distance, where the interaction between fish bodies may play a role. The side-by-side inphase formations trivially satisfy this linearity because $d/UT \approx \phi/2\pi = 0$, but the side-by-side antiphase formations don't satisfy; in the latter, $d/UT = 0$ while $\phi/2\pi = 1$.

These findings strongly indicate that flow-coupled flapping swimmers passively organize into stable traveling equilibrium formations with linear phase-distance relationship. This relationship is independent of the geometric layout (inline versus laterally offset swimmers), flapping kinematics (heaving versus pitching), material properties (rigid versus flexible), tank geometry (rotational versus translational), fidelity of the fluid model (CFD versus VS versus particle model), and system (biological versus robotic, 2D versus 3D). Observations that are robust across such a broad range of systems

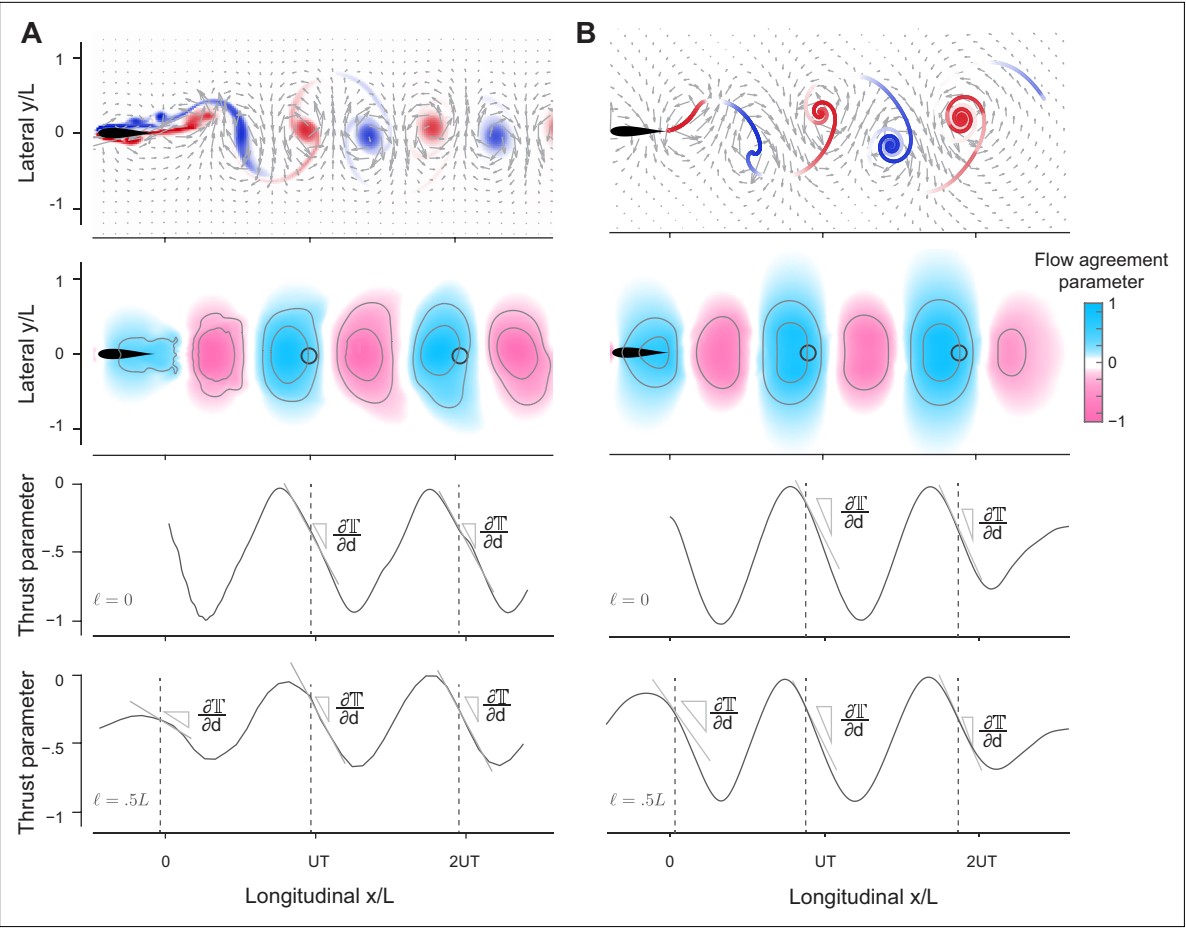

**Figure 4.** Prediction of relative equilibria in the wake of a solitary swimmer. (**A, B**) Snapshots of vorticity and fluid velocity fields created by a solitary swimmer in computational fluid dynamics (CFD) and vortex sheet (VS) simulations and corresponding flow agreement parameter $\mathbb{V}$ fields for a virtual follower at $\phi = 0$. Locations of maximum $\mathbb{V}$ values (i.e. peaks in the flow agreement parameter field) coincide with the emergent equilibria in inphase pairwise formations (indicated by black circles). Contour lines represent flow agreement parameter at ±0.25, ±0.5. Thrust parameter $\mathbb{T}$ is shown at $\ell = 0$ and $\ell = 0.5L$. A negative slope $\partial\mathbb{T}/\partial d$ indicates stability of the predicted equilibria. See also *Figure 1—figure supplements 1 and 2*.

are expected to have common physical and mechanistic roots that transcend the particular set-up or system realization.

Importantly, this universal relationship indicates that flow physics passively positions a swimmer at locations $d$ where the swimmer's flapping phase $\phi$ matches the local phase of the wake $\phi_{\text{wake}} = 2\pi d/UT$, such that the effective phase $\phi_{\text{eff}} = \phi - \phi_{\text{wake}}$ is zero. Importantly, because the quantity $UT$ is nearly equal to the wavelength of the wake of a solitary swimmer, the phase $\phi_{\text{wake}} = 2\pi d/UT$ is practically equal to the phase of a solitary leader. These observations have two major implications. First, they are consistent with the vortex phase matching introduced in *Li et al., 2020*, as a strategy by which fish maximize hydrodynamic benefits. However, they proffer that vortex phase matching is an outcome of passive flow interactions among flapping swimmers, and not necessarily an active strategy implemented by fish via sensing and feedback mechanisms. Second, they led us to hypothesize that emergent side-by-side formations can be predicted from symmetry arguments, while emergent inline and diagonal formations can be predicted entirely from kinematic considerations of the leader's wake without considering two-way flow coupling between the two swimmers.

## Leader's wake unveils opportunities for stable emergent formations

To challenge our hypothesis that the leader's wake contains information about the emergent pairwise equilibria, we examined the wake of a solitary swimmer in CFD and VS simulations (*Figure 4A and B*). By analyzing the wake of a solitary swimmer, without consideration of two-way coupling with a trailing swimmer, we aimed to assess the opportunities available in that wake for a potential swimmer,

undergoing flapping motions, to position itself passively in the oncoming wake and extract hydrodynamic benefit.

Therefore, in the following analysis, we treated the potential swimmer as a 'virtual' particle located at a point $(x, y)$ in the oncoming wake and undergoing prescribed transverse oscillations $A \sin(2\pi ft - \phi)$ in the $y$-direction, at velocity $\mathbf{v}(t; \phi) = 2\pi A f \cos(2\pi ft - \phi)\mathbf{e}_y$, where $\mathbf{e}_y$ is a unit vector in the $y$-direction. The oncoming wake is blind to the existence of the virtual particle. Guided by our previous findings that stable equilibrium formations in pairwise interactions occur at zero effective phase $\phi_{\text{eff}} = \phi - \phi_{\text{wake}} = 0$, where the net hydrodynamic force on the trailing swimmer is zero and where small perturbations lead to negative force gradients, we introduced two assessment tools: a *flow agreement parameter* field $\mathbb{V}(x, y; \phi)$ that measures the degree of alignment, or *matching*, between the flapping motion of the virtual particle and the transverse flow of the oncoming wake, and a *thrust parameter* field $\mathbb{T}(x, y; \phi)$ that estimates the potential thrust force required to undergo such flapping motions.

Specifically, inspired by *Arranz et al., 2022*, and following *Heydari and Kanso, 2021*, we defined the *flow agreement parameter* $\mathbb{V}(x, y; \phi)$ using $\frac{1}{T} \int_t^{t+T} \mathbf{v} \cdot \mathbf{u} \, dt'$, where $t$ is chosen after the oncoming wake has reached steady state, normalized by $\frac{1}{T} \int_t^{t+T} \mathbf{v} \cdot \mathbf{v} \, dt'$ (Appendix E). The normalized $\mathbb{V}(x, y; \phi)$ describes how well the oscillatory motion $\mathbf{v}(t; \phi)$ of the virtual particle matches the local transverse velocity $\mathbf{u}(x, y, t)$ of the oncoming wake (*Heydari and Kanso, 2021*). Positive (negative) values of $\mathbb{V}$ indicate that the flow at $(x, y)$ is favorable (unfavorable) to the flapping motion of the virtual follower.

In *Figure 4A and B*, we show $\mathbb{V}(x, y; \phi = 0)$ as a field over the physical space $(x, y)$ for $\phi = 0$. Blue regions indicate where the local flow favors the follower's flapping motion. In both CFD and VS simulations, the locations with the maximum flow agreement parameter closely coincide with the stable equilibria (black circles) obtained from solving pairwise interactions. These findings imply that hydrodynamic coupling in pairs of flapping swimmers is primarily *non-reciprocal* – captured solely by consideration of the effects of the leader's wake on the follower. This non-reciprocity allows one, in principle, to efficiently and quickly identify opportunities for hydrodynamic benefits in the leader's wake, without the need to perform costly two-way coupled simulations and experiments.

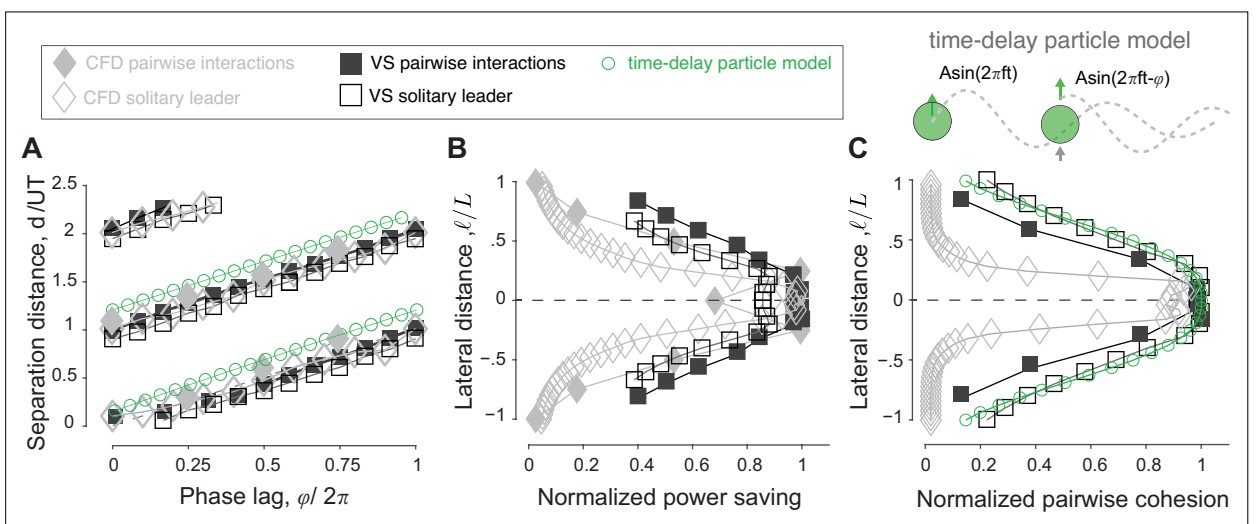

**Figure 5.** Prediction of energetically beneficial, stable equilibria in the wake of a solitary swimmer at various phase lags and lateral offsets. (**A**) Location of maximum $\mathbb{V}$ as a function of phase lag $\phi$ in the wake of solitary leaders in computational fluid dynamics (CFD) and vortex sheet (VS) simulations. For comparison, equilibrium distances of pairwise simulations in CFD, VS, and time-delay particle models (*Figure 5—figure supplement 1*) are superimposed. Agreement between $\mathbb{V}$-based predictions and actual pairwise equilibria is remarkable. (**B**) $\mathbb{V}$ values also indicate the potential benefits of these equilibria, here shown as a function of lateral distance $\ell$ for a virtual inphase follower in the wake of a solitary leader in CFD and VS simulations. The power savings of an actual follower in pairwise formations in CFD and VS simulations are superimposed. (**C**) A negative slope $\partial \mathbb{T}/\partial d$ of the thrust parameter $\mathbb{T}$ indicates stability and $|\partial \mathbb{T}/\partial d|$ expresses the degree of cohesion of the predicted formations, here, shown as a function of $\ell$ for an inphase virtual follower. $|\partial F/\partial d|$ obtained from pairwise formations in VS and time-delay particle models are superimposed (*Figure 5—figure supplement 1*). Results in (**B** and **C**) are normalized by the corresponding maximum values to facilitate comparison.

The online version of this article includes the following figure supplement(s) for figure 5:

**Figure supplement 1.** Time-delay particle model.

Importantly, our findings suggest a simple rule for identifying the locations of stable equilibria in any oncoming wake from considerations of the flow field of the wake itself: a potential swimmer undergoing a flapping motion at phase $\phi$ tends to position itself at locations $(x^*, y^*)$ of maximum flow agreement $\mathbb{V}(x, y; \phi)$ between its flapping motion and the oncoming wake.

To verify this proposition, we show in **Figure 5A**, as a function of phase $\phi$, the streamwise locations of the local maxima of $\mathbb{V}(x, y; \phi)$ computed based on the CFD and VS models, and scaled by $UT$, where $U$ is the speed of the solitary swimmer. We superimpose onto these results the equilibrium configurations obtained from pairwise interactions in the context of the CFD (◆), VS (■), and time-delay particle (◯) models, where we modified the latter to account for non-zero lateral offset $\ell$ (Appendix D and **Figure 5—figure supplement 1**). Predictions of the equilibrium configurations based on maximal flow agreement parameter agree remarkably well with actual equilibria based on pairwise interactions, and they all follow the universal linear phase-distance relationship shown in **Figure 2B**.

The wake of a solitary swimmer contains additional information that allows us to evaluate the relative power savings of a potential follower and relative stability of the pairwise formation directly from the leader's wake, without accounting for pairwise interactions. Assessment of the relative power savings follows directly from the maximal value of the flow agreement parameter: larger values imply more power savings and reduced cost of transport. To verify this, we calculated the maximal $\mathbb{V}(x^*, y^*; \phi)$ in the wake of the solitary swimmer, where we expected the follower to position itself in pairwise interactions. In **Figure 5B**, we plotted these $\mathbb{V}$ values as a function of lateral distance $\ell$ for $\phi = 0$. We superimposed the power savings $\Delta P$ based on pairwise interactions of inphase swimmers using the CFD and VS simulations and normalized all quantities by the maximal value of the corresponding model to highlight variations in these quantities as opposed to absolute values. Power savings are almost constant for $\ell < 0.25L$, but decrease sharply as $\ell$ increases. This trend is consistent across all models, with the most pronounced drop in the CFD-based simulations because the corresponding velocity field $\mathbf{u}$ decays more sharply when moving laterally away from the swimmer.

Next, to assess the stability of the virtual particle based only on information in the oncoming wake of a solitary swimmer, we estimated the thrust force based on the fact that the thrust magnitude scales with the square of the swimmer's lateral velocity relative to the surrounding fluid's velocity (**Triantafyllou et al., 1993**; **Floryan et al., 2017**; **Newbolt et al., 2019**). We defined the thrust parameter field $\mathbb{T}(x, y; \phi) = -\frac{1}{T} \int_t^{t+T} \left| (\mathbf{v} - \mathbf{u}).\mathbf{e}_y \right|^2 dt'$, normalized using $\frac{1}{T} \int_t^{t+T} \left| \mathbf{v}.\mathbf{e}_y \right|^2 dt'$. At the locations of the maxima of $\mathbb{V}(x^*, y^*; \phi)$, a negative slope $\partial \mathbb{T}/\partial d$ of the thrust parameter is an indicator of linear stability or cohesion of the potential equilibria, i.e., emergent pairwise formations are expected to be stable if a small perturbation in distance about the locations $(x^*, y^*)$ of maximal $\mathbb{V}$ is accompanied by an opposite, restorative change in $\mathbb{T}$. Indeed, in both CFD and VS wakes, $\partial \mathbb{T}/\partial d$ at $(x^*, y^*)$ is negative (**Figure 4**).

In **Figure 5C**, we plotted $|\partial \mathbb{T}/\partial d|$ as a function of lateral distance $\ell$ for $\phi = 0$. We superimposed the magnitude of the eigenvalues $|\delta F/\delta d|$ obtained from the linear stability analysis of pairwise interactions in inphase swimmers using the VS and time-delay particle models. As in **Figure 5B**, all quantities are normalized by the maximal value of the corresponding model to highlight variations in these quantities as opposed to absolute values. Also, as in **Figure 5B**, all models produce consistent results: pairwise cohesion is strongest for $\ell < 0.25L$, but weakens sharply as $\ell$ increases, with the most pronounced drop in the CFD-based simulations.

A few comments on our virtual particle model and diagnostic tools in terms of the flow agreement and thrust parameters are in order. Our model differs from the minimal particle model used in **Becker et al., 2015**; **Newbolt et al., 2019**, which treated both swimmers as particles with minimal 'wakes' and considered two-way coupling between them (see Appendix D).

In our analysis, the oncoming wake can be described to any desired degree of fidelity of the fluid model, including using experimentally constructed flows when available. Indeed, our flow agreement and thrust parameters are agnostic to how the flow field of the oncoming wake is constructed. Additionally, these diagnostic tools are equally applicable to any oncoming wake, not necessarily produced by a single swimmer, but say by multiple swimmers (as discussed later) or even non-swimming flow sources. Thus, the approach we developed here could be applied broadly to analyze, predict, and test opportunities for schooling and hydrodynamic benefits for live and robotic fish whenever measurements of an oncoming flow field are available.

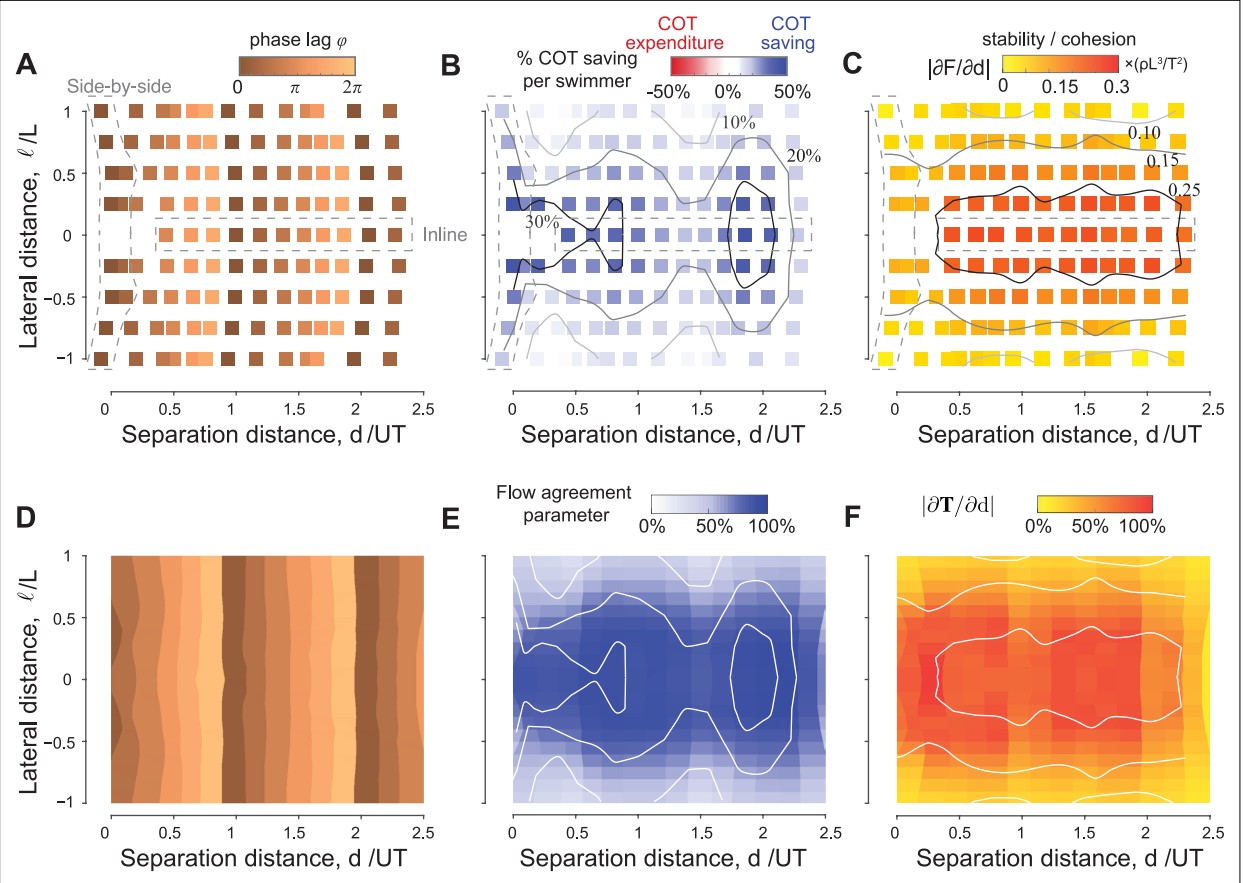

**Figure 6.** Equilibria are dense over the parameter space of phase lags and lateral offsets. For any given phase lag $\phi$ and at any lateral offset $\ell$ inside the wake, the pair reach equilibrium formations that are stable and power saving relative to a solitary swimmer. (**A**) Equilibrium separation distances, (**B**) average power saving, and (**C**) stability as a function of phase lag and lateral distance in a pair of swimmers. Predictions of (**D**) equilibrium locations, (**E**) hydrodynamic benefits, and (**F**) cohesion based on the wake of a solitary swimmer following the approach in *Figures 4 and 5*. For comparison, the contour lines from (B and C) based on pairwise interactions are superimposed onto panels (E and F) (white lines). Simulations in (A–C) are based on pairwise interactions and simulations in (D–F) are based on the wake of a single swimmer, all in the context of the vortex sheet model with $A = 15°$, $f = 1$, and $\tau_{\text{diss}} = 2.45T$.

## Parametric analysis over the entire space of phase lags and lateral offsets

Having demonstrated consistency in the emergence of flow-mediated equilibria in both CFD and VS simulations, we next exploited the computational efficiency of the VS model to systematically investigate emergent pairwise formations over the entire space of phase lag $\phi \in [0, 2\pi)$ and lateral offset $\ell \in [-L, L]$, excluding side-by-side antiphase formations. Equilibrium configurations are dense over the entire range of parameters: for any combination of phase lag $\phi$ and lateral offset $\ell$, there exists an emergent equilibrium configuration where the pair of swimmers travel together at a separation distance $d/UT$ (*Figure 6A*). Perturbing one or both parameters, beyond the limits of linear stability, causes the swimmers to stably and smoothly transition to another equilibrium at different spacing $d/UT$. Importantly, increasing the phase lag $\phi$ shifts the equilibrium positions in the streamwise direction such that $d/UT$ depends linearly on $\phi$, but the effect of lateral distance for $\ell \leq L$ is nonlinear and nearly negligible for small $\ell$: increasing the lateral offset $\ell$ by an entire body length $L$ changes the pairwise distance $d/UT$ by about 15%. Our results explored emergent equilibria up to $d/UT \leq 2.5$ and are consistent with the experimental findings in *Newbolt et al., 2022*, which explored up to nine downstream equilibria.

To assess the hydrodynamic advantages of these emergent formations, we calculated the average change in hydrodynamic power per swimmer. The pair saves power compared to solitary swimming (*Figure 6B*). Power savings vary depending on phase lag $\phi$ and lateral distance $\ell$: for the entire range

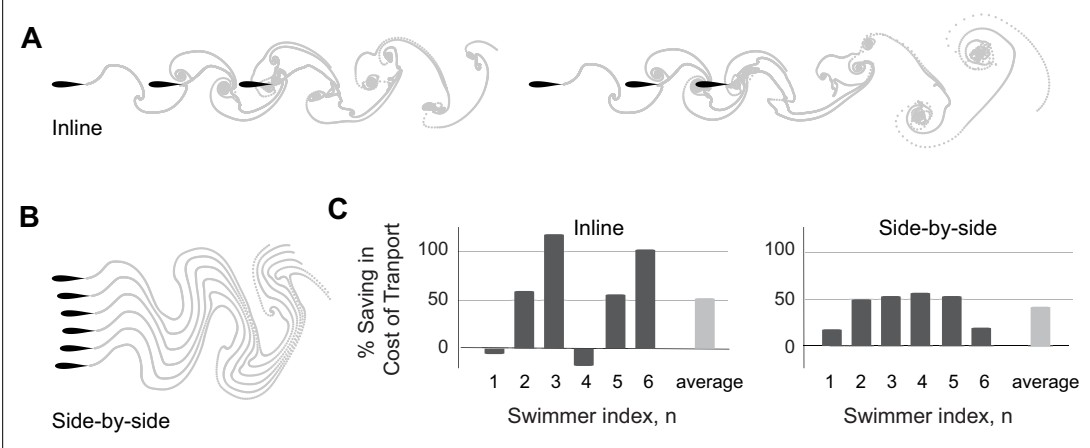

**Figure 7.** Larger inline and side-by-side formations. (**A**) Inline formations lose cohesion and split into two subgroups as depicted here for a group of six swimmers. (**B**) Side-by-side formations remain cohesive. (**C**) Power saving of each swimmer in inline and side-by-side formations. Dissipation time $\tau_{\text{diss}} = 2.45T$. Simulations of inline formations and side-by-side formations ranging from two to six swimmers are shown in *Figure 7—figure supplements 1 and 2*.

The online version of this article includes the following figure supplement(s) for figure 7:

**Figure supplement 1.** Inline formations of multiple flapping swimmers.

**Figure supplement 2.** Side-by-side formations of multiple flapping swimmers.

of $\phi$ from 0 to $2\pi$, the school consistently achieves over 20% power reduction, as long as the lateral offset is $\ell \leq 0.25L$. However, increasing $\ell$ from $0.25L$ to $L$ reduces significantly the hydrodynamic benefit. That is, swimmers can take great liberty in changing their phase without compromising much the average energy savings of the school, as long as they maintain close lateral distance to their neighbor.

A calculation of the linear stability of each equilibrium in *Figure 6A* shows that these emergent formations are linearly stable (*Figure 6C*), and the degree of stability is largely insensitive to phase lag, with strongest cohesion achieved at lateral offset $\ell \leq 0.25L$. The results in *Figure 6A–C* are constructed using pairwise interactions in VS simulations, but can be inferred directly from the wake of a solitary leader, as discussed in the previous section and shown in *Figure 6D–F*.

## Analysis of larger groups of inline and side-by-side swimmers

How do these insights scale to larger groups? To address this question, we systematically increased the number of swimmers and computed the emergent behavior in larger groups based on flow-coupled VS simulations.

In a group of six swimmers, all free to move in the streamwise $x$-direction, we found that the last three swimmers split and form a separate subgroup (*Figure 7A*). In each subgroup, swimmer 3 experiences the largest hydrodynamic advantage (up to 120% power saving!), swimmer 2 receives benefits comparable to those it received in pairwise formation (65% power saving), and swimmer 1 no benefit at all (*Figure 7C*).

We asked if loss of cohesion is dependent on the number of inline swimmers. To address this question, we gradually increased the number of swimmers from two to six (*Figure 7—figure supplement 1*). We found that in a school of three inline swimmers, flow interactions led to a stable emergent formation with hydrodynamic benefits similar to those experienced by the three swimmers in each subgroup of *Figure 7A and C*. When computing the motion of four inline swimmers (*Figure 8A*), we found that the leading three swimmers maintained cohesion, at hydrodynamic benefits similar to a formation of three, but swimmer 4 separated and lagged behind, receiving no advantage in terms of power savings because it split from the formation (*Figure 8D*, *Figure 7—figure supplement 1*). In a group of five, the last two swimmers split and formed their own subgroup. That is, in all examples, swimmer 4 consistently lost hydrodynamic advantage and served as local leader of the trailing subgroup. These observations are consistent with *Peng et al., 2018*, and demonstrate that flow interactions alone are insufficient to maintain inline formations as the group size increases.

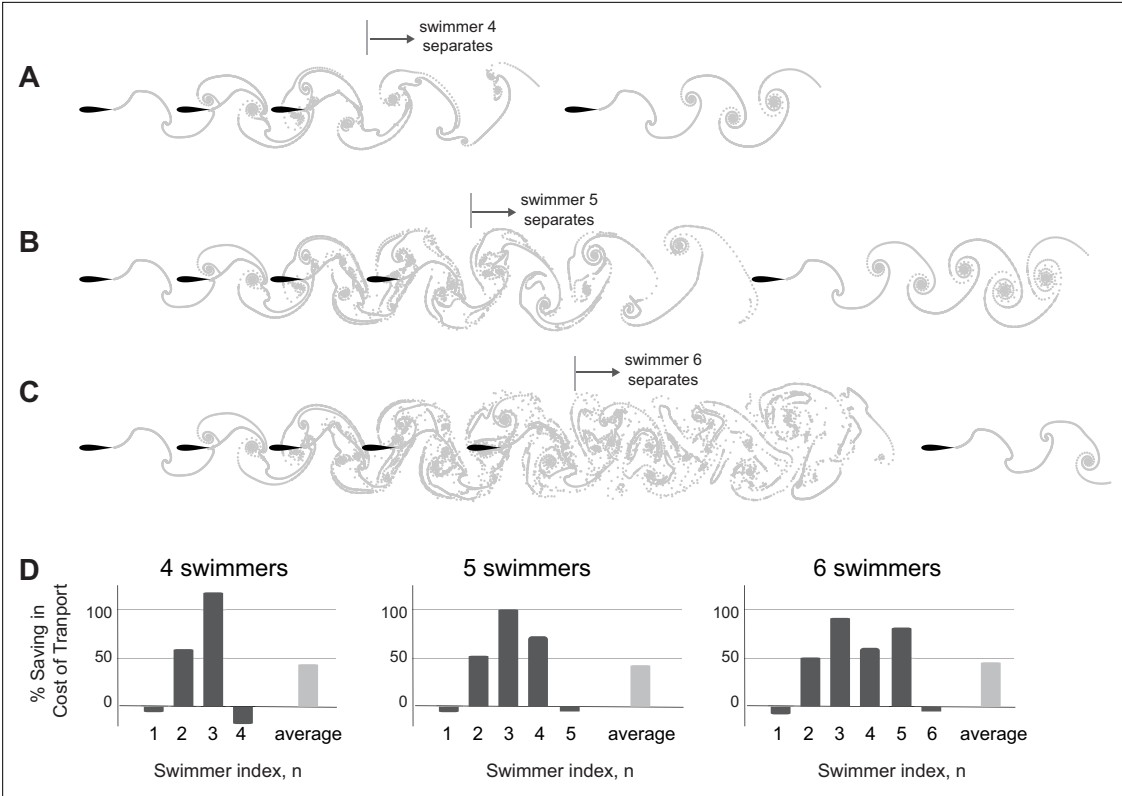

**Figure 8.** Loss of cohesion in larger groups of inline swimmers. Number of swimmers that stay in cohesive formation depends on parameter values. (**A–C**) For dissipation time $\tau_{\mathrm{diss}} = 2.45T$, $3.45T$, and $4.45T$, the fourth, fifth, and sixth swimmers separate from the group, respectively. (**D**) Power savings per swimmer in (A–C), respectively. On average, all schools save equally in cost of transport, but the distribution of these savings vary significantly between swimmers. In all case, swimmer 3 receives the most hydrodynamic benefits.

We next explored the robustness of the side-by-side pattern to larger number of swimmers starting from side-by-side initial conditions (*Figure 7B*). The swimmers reached stable side-by-side formations reminiscent of the configurations observed experimentally when fish were challenged to swim at higher swimming speeds (*Ashraf et al., 2017*). The swimmers in this configuration saved power compared to solitary swimming (*Figure 7C*): swimmers gained equally in terms of hydrodynamic advantage (up to 55% power saving for the middle swimmers in a school of six), except the two edge swimmers which benefited less. We tested these results by gradually increasing the number of swimmers from two to six (*Figure 7—figure supplement 2*). The robustness and overall trend of power saving among group members is robust to the total number of swimmers in these side-by-side formations.

## Mechanisms leading to loss of cohesion in larger inline formations

To understand why three swimmers form a stable inline formation but four don't, we extended the analysis in *Figure 4* to analyze the wake created behind two-swimmer (*Figure 9A*) and three-swimmer (*Figure 9B*) groups. Specifically, we computed pairwise interactions in a two-swimmer school and considered the combined wake of both swimmers after they had settled onto an equilibrium state. Similarly, we computed the behavior of a three-swimmer school and analyzed the combined wake at steady state. Compared to the single leader wake in *Figure 4B*, in the wake of a two-swimmer school, positive flow agreement in the (blue) region is enlarged and enhanced, corresponding to swimmer 3 receiving the largest power savings. On the other hand, behind three inline swimmers, the region of positive flow agreement is weakened and shrunk, indicating weaker potential for energy saving by a fourth swimmer.

Importantly, in the wake of the pairwise formation, the downstream jet is modest at the location of maximum $\mathbb{V}$, where swimmer 3 is expected to position itself for hydrodynamic benefit, thus allowing swimmer 3 to reach this position and stay in formation (*Figure 9E*). Also, at this location, the wake has a substantial transverse velocity $\mathbf{u} \cdot \mathbf{e}_y$ (*Figure 9G*), which aids thrust production at a diminished

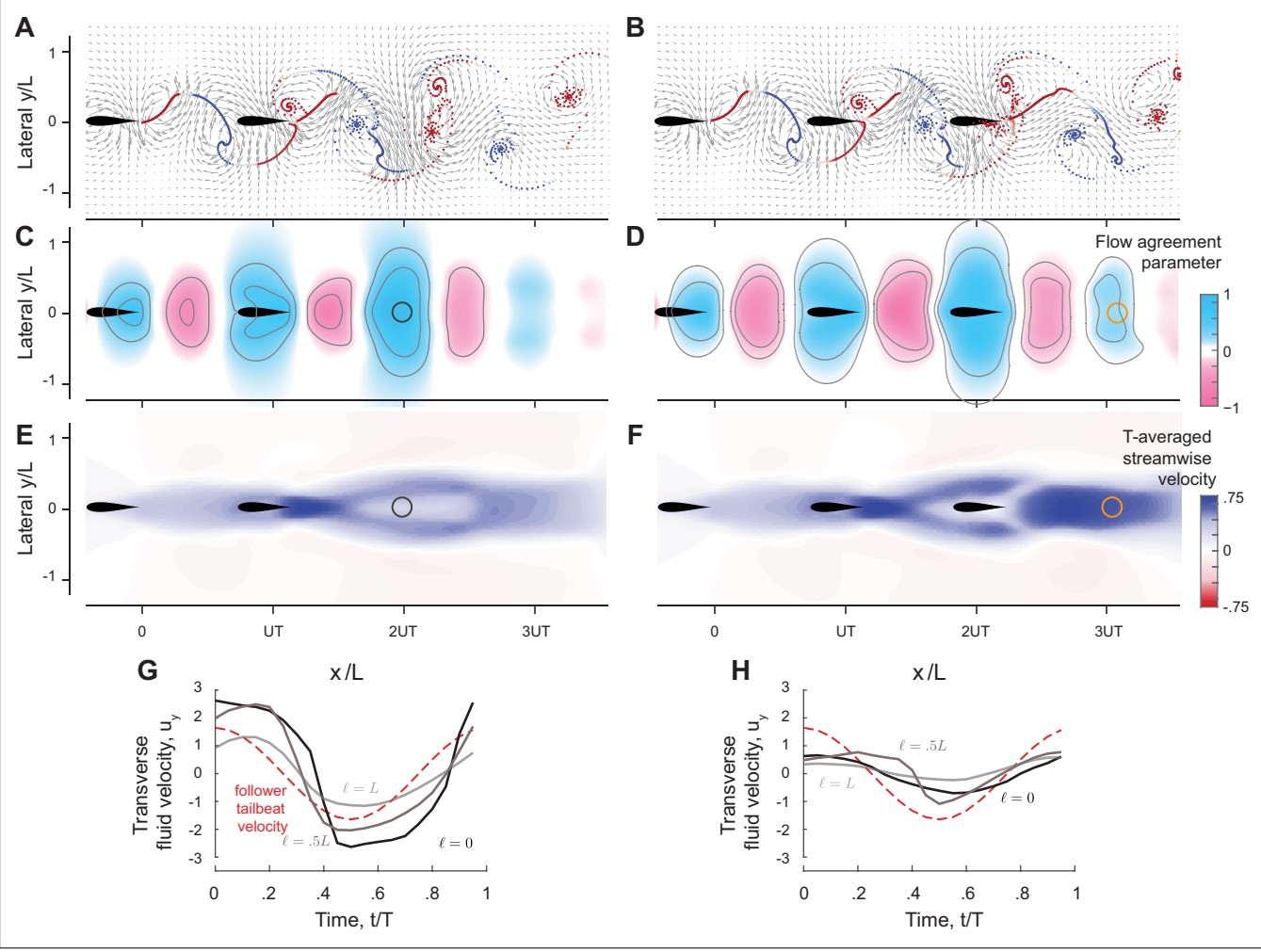

**Figure 9.** Prediction of equilibrium locations in the wake of multiple upstream swimmers. (A, **B**) Snapshots of vorticity fields created by two inline inphase swimmers, and three inline inphase swimmers. (**C**, **D**) show the corresponding flow agreement parameter $\mathbb{V}$ fields. Contour lines represent flow agreement parameter at ±0.25, ±0.5. (**E**, **F**) plot the corresponding period-averaged streamwise velocity. Separation distances $d/UT$ predicted by the locations of maximal $\mathbb{V}$ are marked by circles in the flow agreement field. In the left column, separation distances $d/UT$ based on freely swimming triplets are marked by black circles and coincide with the locations of maximal $\mathbb{V}$. In the right column, the orange marker shows the prediction of the location of a fourth swimmer based on the maximum flow agreement parameter. In two-way coupled simulation, swimmer 4 actually separates from the leading three swimmers as illustrated in *Figure 8A*. Computational fluid dynamics (CFD) simulation shows swimmer 4 will collide with swimmer 3 as in *Figure 9—figure supplement 1*. (**G**, **H**) show the transverse flow velocity in a period at the location predicted by the maximum flow agreement parameter and with a lateral offset $\ell = 0, 0.5L, L$, in comparison to the follower's tailbeat velocity.

The online version of this article includes the following figure supplement(s) for figure 9:

**Figure supplement 1.** Inline formations of multiple flapping swimmers in computational fluid dynamics (CFD) simulations.

cost. In contrast, three inline swimmers generate a much stronger downstream jet at the location of maximum $\mathbb{V}$ where swimmer 4 is expected to position itself (*Figure 9F*). This jet prevents swimmer 4 from stably staying in formation, and the transverse flow velocity $\mathbf{u} \cdot \mathbf{e}_y$ is nearly zero for the entire flapping period (*Figure 9H*), indicating little opportunity for exploiting the flow generated by the three upstream swimmers for thrust generation. This limitation is fundamental; it results from the flow physics that govern the wake generated by the upstream swimmers. There is not much that a trailing swimmer can do to extract hydrodynamic benefits from an oncoming flow field that does not offer any.

## Critical size of inline formations beyond which cohesion is lost

We sought to understand what determines the critical group size, here three, beyond which inline formations lose cohesion and split into subgroups. Because we have established that the flow

agreement parameter $\mathbb{V}$ plays an important role in predicting emergent formations, we first examined $\mathbb{V}$ in the wake of a pair of flapping swimmers in CFD (*Figure 1—figure supplement 1*) and VS (*Figure 1—figure supplement 2*) simulations. These results show that at lower Re and smaller dissipation time $\tau_{\mathrm{diss}}$, the flow agreement parameter $\mathbb{V}$ decays rapidly downstream of the flapping swimmers, thus diminishing the opportunities for downstream swimmers to passively stay in cohesive formation and achieve hydrodynamic benefits. We thus hypothesized that the number of swimmers that passively maintain a cohesive inline formation is not a universal property of the flow physics, but depends on the flow regime.

We tested this hypothesis in VS simulations with increasing number of swimmers and increasing $\tau_{\mathrm{diss}}$. As we increased $\tau_{\mathrm{diss}}$, the number of swimmers that stayed in cohesive inline formation increased (*Figure 8B and C*). These findings confirm that this aspect of schooling – the maximal number of swimmers that passively maintain a cohesive inline formation – is indeed scale-dependent. Interestingly, an analysis of the power savings in these formations shows that, although swimmers 4 and 5 stay in formation at increased $\tau_{\mathrm{diss}}$, swimmer 3 always receives the most hydrodynamic benefit (*Figure 8D*).

We additionally tested the stability of inline formations in CFD simulations at Re=1645 (*Figure 9—figure supplement 1*) and observed the same trend: an inline school of three swimmers remains cohesive, but a fourth swimmer collides with the upstream swimmer. These observations imply that the loss of cohesion does not depend on the specific fluid model. This is consistent qualitatively with existing results (*Peng et al., 2018*). In *Peng et al., 2018*, the authors employed flexible heaving foils at Re = 200 and observed stable inline formations with larger number of swimmers. The flexible foil model and smaller Re make the swimmer more adaptive to changes in the flow field, by passively modulating the amplitude and phase along its body, thus diverting some of the hydrodynamic energy into elastic energy and stabilizing the larger inline formation. This, again, emphasizes that the number of swimmers in a stable inline group is not a universal property of the formation, rather it is model and scale-dependent.

## Mapping emergent spatial patterns to energetic benefits

We next returned to the school of four swimmers, which, when positioned inline and flapped inphase, lost cohesion as the trailing swimmer separated from the school. We aimed to investigate strategies for stabilizing the emergent school formation and mapping the location of each member in the school to the potential benefit or cost it experiences compared to solitary swimming.

Inspired by vortex phase matching as an active strategy for schooling (*Li et al., 2020*; *Li et al., 2021*), we tested whether phase control is a viable approach to maintain cohesion and gain hydrodynamic benefits. We devised an active feedback control strategy, where the swimmer senses the oncoming transverse flow velocity at its location and adjusts its flapping phase to maximize the agreement $\mathbb{V}$ between its flapping motion and the local flow (see Appendix F for more details). When applied to swimmer 4 (*Figure 10A*), this phase controller led to a stable formation, albeit at no benefit to swimmer 4; in fact, swimmer 4 spent 100% more power compared to solitary swimming, whereas the power savings of swimmers 2 and 3 remained robustly at the same values as in the formation without swimmer 4. The inability of swimmer 4 to extract hydrodynamic benefits from the oncoming flow is due to a fundamental physical limitations, as explained in *Figure 9*; by the non-reciprocal nature of flow interactions, changing the phase of the trailing swimmer has little effect on the oncoming flow field generated by the upstream swimmers. If the oncoming wake itself presents no opportunity for hydrodynamic benefit, phase control cannot generate such benefit.

We next investigated whether collaborative phase modulation could aid in maintaining school cohesion by imposing that each swimmer flaps at a phase lag $\Delta\phi$ relative to the swimmer ahead (*Figure 10B and C*). We found a range of values of $\Delta\phi$ at which the school became passively stable, but without providing much hydrodynamic benefit to the trailing swimmer; in fact, at certain $\Delta\phi$, cohesion came at a hydrodynamic cost to swimmer 4, much like the active phase control strategy.

Lastly, we investigated whether lateral offset of some of the swimmers could passively stabilize the emergent formation. The choice of which swimmers to displace laterally and by how much is not unique. Thus, we probed different scenarios and obtained multiple stable formations (*Figure 10D* and *Figure 10—figure supplement 1*, *Video 2*). For example, pairing any two of the four swimmers side-by-side, say at the leading, middle, or trailing end of the school, led to cohesive formations. The distribution of hydrodynamic cost or benefit varied depending on the spatial pattern of the school and

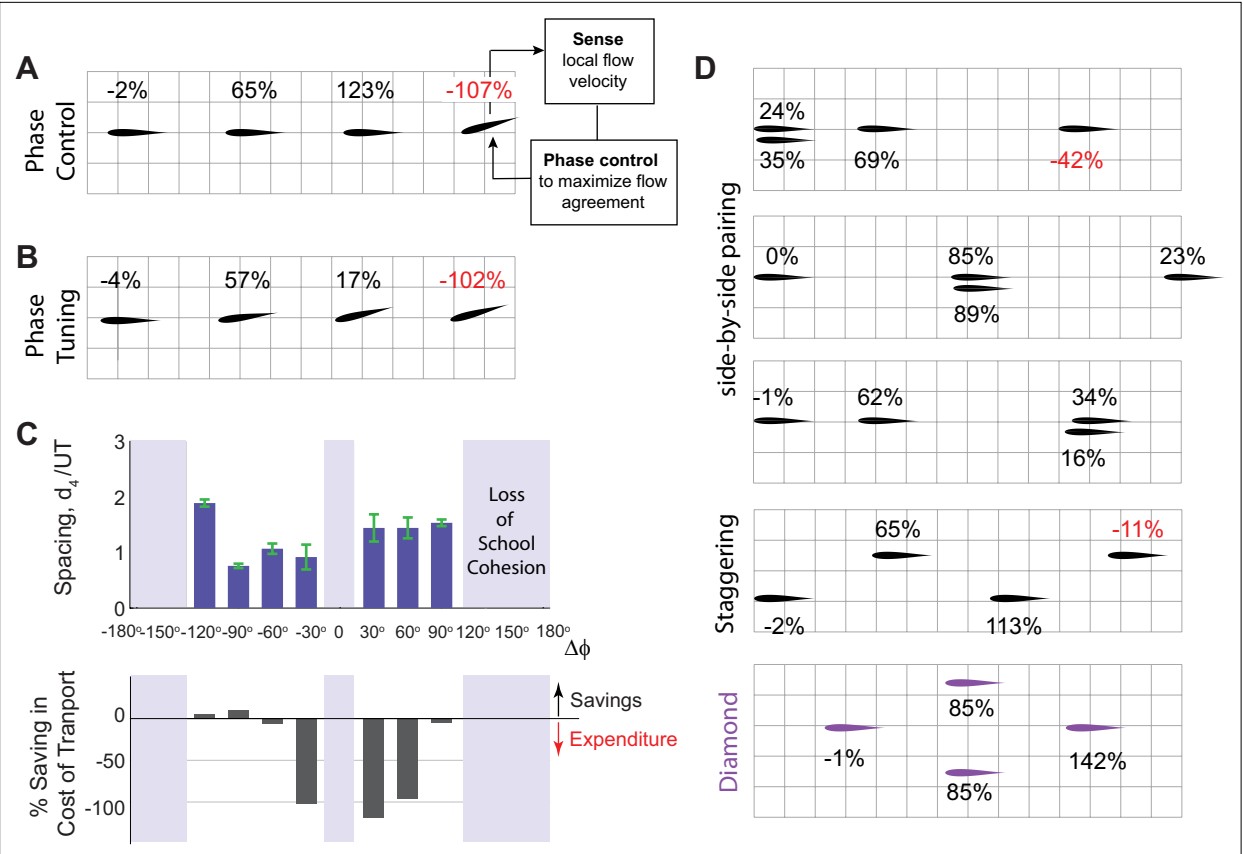

**Figure 10.** Passive and active methods for stabilizing an emergent formation of four swimmers. (**A**) In an inline school of four swimmers, the leading three swimmers flap inphase, but swimmer 4 actively controls its phase in response to the flow it perceives locally to match its phase to that of the local flow as proposed in *Li et al., 2020*. The phase controller stabilizes swimmer 4 in formation but at no hydrodynamic benefit. (**B**) Sequentially increasing the phase lag by a fixed amount $\Delta\phi = -30^o$ in an inline school of four swimmers stabilizes the trailing swimmer but at no hydrodynamic benefit. (**C**) Gradually tuning the phase lag $\Delta\phi$ in a school of four swimmers as done in (B). At moderate phase lags, the school stays cohesive (top plot) but swimmer 4 barely gets any power savings (bottom plot). (**D**) By laterally offsetting the swimmers, four swimmers, all flapping inphase, form cohesive schools with different patterns, e.g., with side-by-side pairing of two swimmers, staggered, and diamond patterns. The time evolution of separation distances is shown in *Figure 10—figure supplement 1*. Individual in each pattern receive a different amount of hydrodynamic benefit. Diamond formation provides the most power saving for the school as anticipated in *Weihs, 1973*, for a school in a regular infinite lattice. In (**A**, **B**, and **D**), %values indicate the additional saving or expenditure in cost of transport relative to solitary swimming.

The online version of this article includes the following figure supplement(s) for figure 10:

**Figure supplement 1.** Alternative formations of four flapping swimmers.

the individual position within the school. Staggering the swimmers in a zigzag pattern also stabilized the school, but did not always allow the trailing swimmer to improve its cost of transport. Staggering the swimmers in a 'diamond' formation stabilized the school and, of all the stable formations we tested, led to the highest savings in cost of transport for the entire school (*Figure 10D*). These results are consistent with existing evidence that diamond formations are both stable (*Tsang and Kanso, 2013*) and energetically optimal (*Weihs, 1973*; *Dai et al., 2018*). But unlike individuals in an infinite diamond lattice (*Weihs, 1973*), individuals in a finite diamond formation do not receive equal energetic benefits.

Our findings highlight the versatility and fluidity of the emergent spatial patterns in groups of flapping swimmers and emphasize that energetic benefits vary depending on the position of the individual within the school. Importantly, these findings imply that, although many emergent formations do not globally optimize the savings of the entire school, hydrodynamic interactions within these formations offer individuals numerous opportunities to achieve varying levels of energetic savings (*Marras and Porfiri, 2012*), potentially creating competition among school members over advantageous positions in the school.

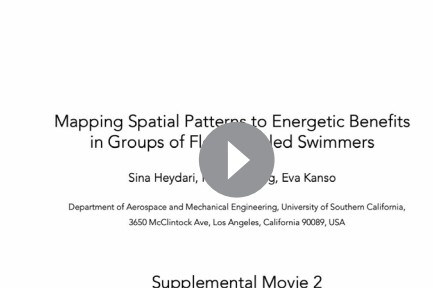

**Video 2.** Groups of flow-coupled swimmers self-organize into inline, side-by-side, or staggered patterns of distinct cohesion and energy-saving properties. Results based on flapping plates in vortex sheet simulations.

https://elifesciences.org/articles/96129/figures#video2

## Discussion

We analyzed how passive flow interactions mediate self-organization in groups of flapping swimmers, all heading in the same direction. Our approach relied on a hierarchy of FSI models and aimed to distill which aspects of self-organization are universal and those which are scale-dependent.

We found that a pair of flapping swimmers self-organize into inline, diagonal, or side-by-side formations (*Figure 1*). The emergent formation depends on the swimmers' flapping phase and initial conditions. In fact, the distinction between these types of formation is somewhat arbitrary because, as phase varies, the emergent equilibria are dense over the space of lateral offset and separation distance (*Figure 6*). These findings are consistent with experimental observations (*Li et al., 2020*; *Thandiackal and Lauder, 2023*; *Newbolt et al., 2019*; *Newbolt et al., 2022*), but go beyond these observations to quantify the hydrodynamic benefits to each member in these formations. Two side-by-side swimmers flapping inphase save energy, compared to solitary swimming, and share the hydrodynamic benefits nearly equally. When flapping antiphase, the side-by-side swimmers exert extra effort compared to solitary swimming, contrary to a common misconception that this configuration saves hydrodynamic energy (*Zhang and Lauder, 2023b*). In leader-follower formations, whether inline or diagonal, hydrodynamic benefits are bestowed entirely on the follower (*Figure 3A*).

Importantly, we showed that the wake of a solitary leader contains information that unveils opportunities for the emergence of stable and energetically favorable formations in pairs of swimmers. Equilibrium locations and trends in power savings and school cohesion can all be predicted entirely from kinematic considerations of the leader's wake with no consideration of the two-way coupling between the two swimmers (*Figure 5*). These results are important because they highlight the non-reciprocal or asymmetric nature of flow coupling in leader-follower configurations, inline or diagonal, at finite Re and open new avenues for future studies of non-reciprocal flow-coupled oscillators. These oscillators have distinct properties from classic mechanical and biological oscillators, such as Huygens pendula or viscosity-dominant oscillators, where the coupling between the oscillators is reciprocal; see, e.g., *Strogatz and Stewart, 1993*; *Lushi et al., 2014*; *Oliveira and Melo, 2015*; *Wan and Goldstein, 2016*; *Guo et al., 2018*.

Our analysis has practical importance in that it provides efficient diagnostics and predictive tools that are equally applicable to computational models and experimental data and could, therefore, be applied broadly to analyze, predict, and test opportunities for schooling and hydrodynamic benefits in live and robotic fish when flow measurements are available.

Case in point, we used these diagnostic tools to explain the mechanisms leading to scattering in larger groups of inline swimmers and to predict when the wake of a leading group of swimmers offers no opportunities for a follower to benefit from passive hydrodynamics (*Figure 9*). At an increasing number of flow-coupled swimmers, side-by-side formations remain robust, but inline formations become unstable beyond a critical number of swimmers (*Figures 7 and 8*). The critical number depends on the fluid properties and can be predicted by analyzing the wake of the leading group of swimmers. Future work will focus on testing these findings experimentally and in CFD simulations with increasing number of swimmers, together with accounting for body deformations (*Hang et al., 2022*), lateral dynamics (*Kurt et al., 2021*; *Das et al., 2023*), and variable flapping amplitudes and frequencies (*Newbolt et al., 2019*; *Hang et al., 2024*).

Our findings could have far-reaching consequences on our understanding of biological fish schools. Field and laboratory experiments (*Partridge and Pitcher, 1979*; *Marras et al., 2015*; *Ashraf et al., 2017*; *Lombana and Porfiri, 2022*) have shown that actual fish schools do not generally conform to highly regularized patterns, and schooling fish dynamically change their position in the school.

Neighboring fish vary from side-by-side to inline and diagonal configurations. Importantly, in laboratory experiments that challenged groups of fish to sustain high swimming speeds, the fish rearranged themselves in a side-by-side pattern as the speed increased, much like the pattern in *Figure 7B*, presumably to save energy (*Ashraf et al., 2017*). These empirical observations, together with our findings that side-by-side formations provide the fairest distribution of efforts among school members (*Figure 7B and C*), offer intriguing interpretations of the results in *Ashraf et al., 2017*; *Lombana and Porfiri, 2022*: when the fish are not challenged by a strong background current to sustain high swimming speeds, they position themselves as they please spatially, without much consideration to equal sharing of hydrodynamic benefits. But when challenged to swim at much higher speeds than their average swimming speed, fish are forced to cooperate.

To expand on this, our results suggest a connection between flow physics and what is traditionally thought of as social traits: greed versus cooperation. We posit that there is a connection between the resources that arise from flow physics – in the form of energetic content of the wake of other swimmers – and greedy versus cooperative group behavior. In cohesive inline formations, the leader is always disadvantaged and hydrodynamic benefits are accorded entirely to trailing swimmers (*Figures 3A and 7C*). Importantly, flows generated by these inline formations present serious impediments for additional swimmers to join the line downstream (*Figures 7 and 8*). Thus, we could call these formations greedy, leaving no resources in the environment for trailing swimmers. This thought, together with our interpretation of the observations in *Ashraf et al., 2017*, that cooperation to achieve an egalitarian distribution of hydrodynamic benefits is forced, not innate, raise an interesting hypothesis. The dynamic repositioning of members within the school (e.g. *Figure 10*) could be driven by greed and competition to occupy hydrodynamically advantageous positions, much like in peloton of racing cyclists (*Blocken et al., 2018*). On a behavioral time scale, these ideas, besides their relevance to schooling fish, open up opportunities for analyzing and comparing the collective flow physics in cooperative versus greedy behavior in animal groups from formations of swimming ducklings (*Yuan et al., 2021*) and flying birds (*Bialek et al., 2012*; *Portugal et al., 2014*) to peloton of racing cyclists (*Blocken et al., 2018*). From an evolutionary perspective, it is particularly exciting to explore the prospect that flow physics could have acted as a selective pressure in the evolution of social traits such as cooperation and greed in aquatic animal groups.

## Acknowledgements

Many thanks to members of the McHenry Laboratory at UC Irvine and Kanso Laboratory at USC for numerous discussions about fish schooling behavior and for comments on the manuscript.

## Additional information

### Funding

| Funder | Grant reference number | Author |
|---|---|---|
| National Science Foundation | IOS-2034043 | Eva Kanso |
| National Science Foundation | CBET-210020 | Eva Kanso |
| Office of Naval Research | N00014-22-1-2655 | Eva Kanso |
| Office of Naval Research | N00014-19-1-2035 | Eva Kanso |

The funders had no role in study design, data collection and interpretation, or the decision to submit the work for publication.

### Author contributions

Sina Heydari, Haotian Hang, Data curation, Software, Investigation, Visualization, Writing – review and editing; Eva Kanso, Conceptualization, Formal analysis, Supervision, Funding acquisition, Investigation, Writing - original draft, Project administration, Writing – review and editing

## Author ORCIDs
Sina Heydari (iD) https://orcid.org/0000-0001-8907-5751
Haotian Hang (iD) https://orcid.org/0000-0001-5217-8124
Eva Kanso (iD) https://orcid.org/0000-0003-0336-585X

Reviewer #1 (Public review): https://doi.org/10.7554/eLife.96129.3.sa1
Reviewer #2 (Public review): https://doi.org/10.7554/eLife.96129.3.sa2
Author response https://doi.org/10.7554/eLife.96129.3.sa3

## Additional files

### Supplementary files
• MDAR checklist

### Data availability
All data is included in the main manuscript and figure supplements. Time evolution data of swimmers obtained from CFD and vortex-sheet simulations are available at https://doi.org/10.5061/dryad.hqbzkh1s5.

The following previously published dataset was used:

| Author(s) | Year | Dataset title | Dataset URL | Database and Identifier |
| --- | --- | --- | --- | --- |
| Heydari S, Hang H, Kanso E | 2024 | Mapping Spatial Patterns to Energetic Benefits in Groups of Flow-coupled Swimmers | https://doi.org/10.5061/dryad.hqbzkh1s5 | Dryad Digital Repository, 10.5061/dryad.hqbzkh1s5 |

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

## Appendix 1

### Computational Fluid Dynamics (CFD) model

In our CFD simulations, a swimmer is modeled as a symmetric 2D Joukowsky airfoil (*Tsien, 1943*). The chord length of the airfoil is the characteristic length $L$, and the maximum thickness is $0.12L$. The airfoil undergoes pitching motion around its leading edge. FSI are governed by the incompressible Navier-Stokes equations,

$$\frac{\partial \mathbf{u}}{\partial t} + \mathbf{u} \cdot \nabla \mathbf{u} = -\nabla p + \frac{1}{\text{Re}}\Delta \mathbf{u}, \qquad \nabla \cdot \mathbf{u} = 0, \tag{2}$$

where $\mathbf{u}(\mathbf{x}, t)$ and $p(\mathbf{x}, t)$ are the velocity and pressure field, respectively. We solved for these fields numerically using IBM that handles the two-way coupled fluid structure interaction (*Peskin, 1977*; *Griffith et al., 2007*; *Griffith and Peskin, 2005*; *Bhalla et al., 2013*; *Griffith and Patankar, 2020*; *Mittal et al., 2008*).

The immersed boundary formulation involves an Eulerian descriptions of the flow field and a Lagrangian description of the immersed swimmers, modeled as Joukowsky airfoils. The boundary condition is mapped to a body force exerted on the fluid. The Lagrangian and Eulerian variables are correlated by the Dirac delta function, which is smoothed during discretization. Here, we used the implementation developed by the group of Prof. Boyce Griffith, *IBAMR, 2024*, which has long been used to solve problems such as blood flow in heart (*Peskin, 1977*; *Lee et al., 2020*), water entry/exit problems (*Bhalla et al., 2020*), fish's swimming (*Hoover and Tytell, 2020*; *Tytell et al., 2014*; *Voesenek et al., 2020*), insect's flight (*Van Buren et al., 2019*; *van Veen et al., 2020*), flexible propulsors (*Hoover et al., 2021*; *Tytell et al., 2016*; *Hoover et al., 2018*), self propulsion of pitching/heaving airfoil (*Hoover et al., 2018*; *Yang and Wu, 2022*), and fish schooling (*Yang and Wu, 2022*; *Lin et al., 2022*). This implementation is based on an adaptive mesh, which enables us to accurately simulate self propulsion and reach steady state in a large computational domain with a reasonable computational cost. The computational domain is a rectangle of dimensions $80L \times 20L$, with periodic boundary conditions on the computational domain and no-slip boundary condition on the surface of airfoils. The initial location of the first swimmer is $12L$ away from the right boundary in streamwise direction. The initial distance between the two swimmers $d(t = 0)$ ranges from $1.5L$ to $4L$ to ensure we access different equilibria that emerge in these pairwise formations. The coarsest Eulerian mesh is a uniform 500×125 Cartesian grid. The computational domain close to the airfoils and their wake are refined. There are three layers of refinement mesh, and the refinement ratio for each layer is 4. The simulation timestep is adaptive, with the maximum timestep $\Delta t_{\max} = 2.5 \times 10^{-3}$.

The hydrodynamic forces $F_x$, $F_y$, and moment $M$ acting on each swimmer are calculated by integrating over the surface of that swimmer, the traction force $\boldsymbol{\sigma} \cdot \mathbf{n}$ and moment $\mathbf{x} \times (\boldsymbol{\sigma} \cdot \mathbf{n})$, where $\boldsymbol{\sigma} = -p\mathbf{I} + \mu(\nabla \mathbf{u} + \nabla \mathbf{u}^T)$ is the fluid stress tensor, $\mathbf{x}$ denotes positions on the surface of that swimmer, and $\mathbf{n}$ the unit normal to that airfoil into the fluid. The hydrodynamic forces and moment are used to solve the equations of motion (*Equation 1*) for each swimmer and obtain the active moment $M_a$ in *Equation 7* needed to evaluate the hydrodynamic power in *Equation 8* that is expended by each swimmer.

## Appendix 2

## Vortex sheet (VS) model

In the VS model, each swimmer is modeled as a rigid plate of length $L$, and it is approximated by a bound vortex sheet, denoted by $l_b$, whose strength ensures that no fluid flows through the rigid plate, and the separated shear layer is approximated by a free regularized vortex sheet $l_w$ at the trailing edge of the swimmer. The total shed circulation $\Gamma$ in the vortex sheet is determined so as to satisfy the Kutta condition at the trailing edge, which is given in terms of the tangential velocity components above and below the bound sheet and ensures that the pressure jump across the sheet vanishes at the trailing edge. To express these concepts mathematically, it is convenient to introduce the complex notation, such that $z = x + iy$, where $i = \sqrt{-1}$. For more details on the VS model and our implementation of it, we refer the readers to appendix A of *Heydari and Kanso, 2021*, as well as to *Nitsche and Krasny, 1994*; *Jones, 2003*; *Sheng et al., 2012*; *Jones and Shelley, 2005*; *Alben and Shelley, 2008*; *Alben, 2009*; *Huang et al., 2016*; *Huang et al., 2018*; *Hang et al., 2022*.

The pressure difference across the infinitely thin swimmers $n[p]_\mp$ is given by integrating the balance of momentum equation for inviscid planar flow along a closed contour containing the vortex sheet and trailing edge. Here, $n = n_x + in_y$ is the unit normal, in complex notation, and $[p]_\mp$ is the jump in pressure across the swimmer. The hydrodynamic force $F = \int_{l_b} n[p]_\mp ds$ acting on the swimmer is given by

$$\int_{l_b} n[p]_\mp ds = F_x + iF_y = -F\sin\theta + iF\cos\theta. \tag{3}$$

The hydrodynamic moment $M$ acting on the swimmer about its leading edge is given by

$$M = \text{Re}\left[\int_{l_b} i\overline{n}(z_{\text{l.e.}} - z_b)[p]_\mp ds\right], \tag{4}$$

where $z_{\text{l.e.}}$ is position of the leading edge $s = 0$. We introduce a drag force $D$ that emulates the effect of skin friction due to fluid viscosity in the context of the VS model (*Fang, 2016*; *Heydari and Kanso, 2021*). Namely, following *Fang, 2016*; *Heydari and Kanso, 2021*, we write the drag force for a swimming plate

$$D = -C_d(\overline{U}_+^{3/2} + \overline{U}_-^{3/2}), \tag{5}$$

where $C_d$ is a drag coefficient and $\overline{U}_\pm$ are the spatially averaged tangential fluid velocities on the upper and lower side of the plate, respectively, relative to the swimming velocity $U$,

$$\overline{U}_\pm(t) = \frac{1}{2l}\int_{-l}^{l} u_\pm(s,t)ds - U, \tag{6}$$

where $u_\mp(s,t)$ denotes the tangential slip velocities on both sides of the plate. We estimate $C_d$ to be approximately 0.05 in the experiments of *Ramananarivo et al., 2016*. Additionally, following *Huang et al., 2016*; *Huang et al., 2018*; *Hang et al., 2022*; *Heydari and Kanso, 2021*, we emulate the effect of viscosity on the wake itself by allowing the shed vortex sheets to decay gradually by dissipating each incremental point vortex after a finite time $\tau_{\text{diss}}$ from the time it is shed into the fluid. Larger $\tau_{\text{diss}}$ implies that the incrementally shed vorticity along the vortex sheet stays in the fluid for longer times, mimicking the effect of lower fluid viscosity. We studied the effect of dissipation time in *Figure 1—figure supplements 2 and 3*, and used $\tau_{\text{diss}} = 2.45T$ in the rest of this paper for the sake of computational cost. Pitching motions are produced by an active moment $M_a$ imposed by the swimmer on the surrounding fluid about the leading edge. The value of $M_a$ is obtained from the balance of angular momentum about the swimmer's leading edge (l.e.),

$$I\ddot{\theta} - \text{Im}[m(\dot{x} + i\dot{y})w_{\text{l.e.}}] = M + M_a, \tag{7}$$

In the VS model, $I = mL^2/3$ is the swimmer's moment of inertia about the leading edge, $w_{\text{l.e.}}$ is the swimmer's velocity at the leading edge (in complex form), and $M$ is the hydrodynamic moment about the leading edge given in *Equation 4*. The hydrodynamic power $P$ expended by the flapping swimmer to maintain its pitching motion is given by

$$P = M_{\text{a}}\dot{\theta}. \tag{8}$$

## Appendix 3

### Time-delay particle model

We supplemented these CFD and VS models by studying pairwise interactions in the context of the minimal particle model used in *Becker et al., 2015*; *Newbolt et al., 2019*. This particle model was designed for inline swimmers. Here, we modify it slightly to account for lateral offset $\ell$ between the swimmers as shown in *Figure 5—figure supplement 1*. In a nutshell, the model assumes that the leader leaves behind a vertical wake speed equal to the leader's flapping speed at the tail. The speed of the 'wake' that is left behind decays exponentially in time, as an approximation for viscous dissipation. Details of the model can be found in the Supplementary Information of *Newbolt et al., 2019*.

In this model, each swimmer is a particle of mass per unit depth $m$ undergoing vertical oscillations such that $y_1 = a\sin(2\pi f t)$ and $y = a\sin(2\pi f t - \phi)$, where $a = L\sin A$ is the tailbeat amplitude of the pitching airfoil. In this model, each particle is assumed to experience a thrust force $F_j$ that is proportional to the square of its vertical velocity relative to the surrounding fluid, and a drag force $D_j$ relative to its relative horizontal speed. Namely,

$$m\ddot{x}_j = -F_j + D_j, \qquad j = 1, 2, \tag{9}$$

where

$$F_j = \rho L C_T(\dot{y}_j - u_y(x_j, y_j))^2, \qquad D_j = \rho L C_D(\dot{x}_j - u_x(x_j, y_j))^2. \tag{10}$$

Here, $u_x$ and $u_y$ are the $x$- and $y$-components of the fluid velocity, $C_T, C_D$ are constant thrust and drag coefficients. Since the leader swims into quiescent fluid, we assume that $u_y(x_1, y_1) = 0$. The follower swims into the wake of the leader, which we assume to have zero horizontal velocity ($u_x = 0$) and vertical velocity that decays exponentially both in time and in the lateral direction,

$$u_y(x_2, y_2) = \dot{y}_1(t - \Delta t)e^{-\Delta t/\tau}e^{-|\ell/h|^p}, \tag{11}$$

where $\Delta t$ is the delay time between the leader and follower, i.e., the time past since the leader passed by the follower's current location: $x_1(t - \Delta t) = x_2(t)$. We added the last term in *Equation 11* to consider decay in the lateral direction $\ell$; to estimate the parameters $p$ and $h$, we used a best curve fit to the data of period-average velocity magnitude versus lateral distance in the wake of a single swimmer in the VS model (*Figure 5—figure supplement 1*). We numerically integrated (*Equation 10*) and solved for the motion of the follower. At steady state, we computed the separation distance $d = x_2 - x_1$ between the pair acting on the follower for a range of $\phi \in [0, 2\pi]$ and $\ell \in [-L, L]$ (*Figure 5—figure supplement 1*). The parameter values are chosen to be consistent with the experiments of Newbolt et al. (provided in Table S1 of SI in *Newbolt et al., 2019*). Specifically, $\rho = 1\,\text{g/cm}^3$, $L = 4\,\text{cm}$, $m = 5.3\,\text{g/cm}$, $C_D = 0.25$, $C_T = 0.96$, $\tau = 0.5\,\text{s}$.

## Appendix 4

### Flow agreement and thrust parameters

The flow agreement parameter field is defined

$$\mathbb{V}(x, y; \phi) = \frac{1}{\int_t^{t+T} \mathbf{v} \cdot \mathbf{v} \, dt'} \left[ \int_t^{t+T} \mathbf{v} \cdot \mathbf{u} \, dt' \right], \tag{12}$$

where $\mathbf{u}(x, y, t)$ is the flow field of an oncoming wake and $\mathbf{v}(t; \phi)$ is flapping velocity of a virtual particle located at a location $(x, y)$ in the oncoming wake. The thrust parameter field is defined as

$$\mathbb{T}(x, y, \phi) = -\frac{1}{\int_t^{t+T} |\mathbf{v} \cdot \mathbf{e}_y|^2 \, dt'} \left[ \int_t^{t+T} |(\mathbf{v} - \mathbf{u}) \cdot \mathbf{e}_y|^2 \, dt' \right]. \tag{13}$$

To further illustrate the meaning of these parameters, we considered an ideal scenario, where the flow velocity is simply of the form of

$$\mathbf{u} = u_x \mathbf{e}_x + u_y \mathbf{e}_y = -U\mathbf{e}_x + 2\pi f A_u \cos(2\pi ft)\mathbf{e}_y, \tag{14}$$

where $u_y(t) = 2\pi f A_u \cos(2\pi ft)$ is the transverse fluid velocity. We placed a virtual or 'ghost' swimmer heaving at a velocity $v(t; \phi) = 2\pi f A_v \cos(2\pi ft - \phi)$. Under this scenario, flow agreement parameter is

$$\mathbb{V} = \frac{A_u}{A_v} \cos \phi \tag{15}$$

and thrust parameter is

$$\mathbb{T} = A_u^2 + A_v^2 - 2A_v A_u \cos \phi \tag{16}$$

Let's now consider a balance of forces on the 'ghost swimmer'. The ghost swimmer is in relative equilibrium if the sum of drag forces $D$ and thrust forces $\mathbb{T}$ are zero,

$$D + \mathbb{T} = 0 \quad \Rightarrow \quad \cos \phi = \frac{A_u^2 + A_v^2 - D}{2A_v A_u} \quad \Rightarrow \quad \phi = \pm \cos^{-1} \frac{A_u^2 + A_v^2 - D}{2A_v A_u}. \tag{17}$$

Substituting back into *Equation 15*, we get

$$\mathbb{V} = \frac{A_u^2 + A_v^2 - D}{2A_v^2} \tag{18}$$

According to our prediction based on the flow agreement parameter, equilibria are located at maximal flow agreement parameter, for which $\phi = 0$ and $\mathbb{V} = A_u/A_v$. This is close to the calculated values of $\phi$ and $\mathbb{V}$ when $A_u \approx A_v$ and $D \approx 0$, i.e., assuming the ghost swimmer is getting a free ride with the local flow at zero thrust and drag.

To consider stability at the calculated equilibria in *Equation 17*, we calculate the derivative of thrust parameter in *Equation 16* with respect to phase $\phi$ and evaluate it at the equilibrium points

$$\frac{\partial \mathbb{T}}{\partial \phi} = -2A_v A_u \sin \phi = \mp 2A_v A_u \sqrt{1 - \frac{A_u^2 + A_v^2 - D^2}{2A_v A_u}} \tag{19}$$

Thus, there is always a stable equilibrium and an unstable equilibrium, both of them are close to the local maximum of flow agreement parameter $\mathbb{V} = A_u/A_v$, which is consistent with *Figure 4* of the main text.

# Appendix 5

## Phase control

Consider a swimmer flapping at a phase $\phi$ can 'sense' or measure the agreement of its flapping motion with the local fluid velocity $\mathbf{u}$ (generated by sources other than itself) at its location (say at its midpoint), over a time span of $m$ flapping periods. The goal of the swimmer would be to adjust its current flapping phase $\phi$ to a desired phase $\Phi$ that maximizes its agreement with the local velocity

$$\Phi(t) = \underset{\phi}{\operatorname{argmax}} \frac{\frac{1}{mT} \int_{\min(t-mT,0)}^{t} \mathbf{v}(t',\phi) \cdot \mathbf{u}(t') \, dt'}{\frac{1}{mT} \int_{\min(t-mT,0)}^{t} \mathbf{v}(t',\phi) \cdot \mathbf{v}(t',\phi) \, dt'}, \tag{20}$$

using a proportional phase controller inspired from *Li et al., 2021*; *Li et al., 2020*,

$$\ddot{\phi}(t) = -\gamma^2 \left[ \phi(t) - \Phi(t) \right] - 2\gamma \dot{\phi}(t); \tag{21}$$

Here, $\Phi(t)$ is the desired phase and $\gamma$ is a constant that determines the speed of convergence. We chose the parameters as follows: we set the number of periods $m = 2$ that describes the memory of the swimmer of the ambient fluid $\mathbf{u}$, such that the time history is two times the pitching period $2T$. We set the control gain $\gamma = 3$ to ensure that the actual phase $\phi(t)$ can reach the desired phase $\Phi(t)$ at 1% of relative error within $1.5T$.

By implementing this phase controller in swimmer 4 in a group of four inline swimmers, the swimmer is able to stabilize itself in the formation as shown in *Figure 10*. However, this stabilization is expensive: by actively controlling its phase to stay in formation, swimmer 4 spends 100% more hydrodynamic power than swimming alone.

## Appendix 6

### Data collection from published literature

To probe the relationship between the scaled separation distance $d/UT$ in pairs of swimmers and their flapping phase lag $\phi$, we collected data from published literature on pairs of interacting swimmers (*Kim et al., 2010*; *Newbolt et al., 2019*; *Li et al., 2020*; *Kurt et al., 2021*; *Heydari and Kanso, 2021*); see *Table 1* and the Supplemental Excel Sheet. We excluded a few studies from this literature survey (*Park and Sung, 2018*; *Dai et al., 2018*), because we were unable to extract $d/UT$ from the data they provided. We also excluded (*Becker et al., 2015*) because their study involved fixed inter-swimmer distances. We found that the relationship between phase lag $\phi$ and separation distance $d/UT$ is approximately linear $\phi/2\pi = d/UT + \beta$. The value of $\beta$ depends on how one defines the inter-swimmer separation distance. The two definitions that are commonly used in the literature are tail-to-tail or tail-to-head (gap) distance. These definitions do not change the linear scaling between $d/UT$ and $\phi/2\pi$; they only change the value of $\beta$. For the data provided in *Figure 3* of the main text, we used the definition of separation distance $d$ that results in $\beta = 0$. Not all literature provided values for $U$ or $T$. In the very few cases when this information were missing, we estimated $UT$ from the linear relationship $\phi/2\pi = d/UT$ (*Kim et al., 2010*).

