## [Editor Report · eLife assessment]

This **fundamental** study provides a modeling regime that provides new insight into the energy-preservation parameters among schooling fish. The strength of the evidence supporting observations such as distilled dynamics between leading and lagging schooling fish which are derived from emergent properties is **compelling**. Overall, the study provides exciting insights into energetic coupling with respect to group swimming dynamics.

---

## [Referee Report · Reviewer #1 (Public review)]

Summary:

The study seeks to establish accurate computational models to explore the role of hydrodynamic interactions on energy savings and spatial patterns in fish schools. Specifically, the authors consider a system of (one degree-of-freedom) flapping airfoils that passively position themselves with respect to the streamwise direction, while oscillating at the same frequency and amplitude, with a given phase lag and at a constant cross-stream distance. By parametrically varying the phase lag and the cross-stream distance, they systematically explore the stability and energy costs of emergent configurations. Computational findings are leveraged to distill insights into universal relationships and clarify the role of the wake of the leading foil.

Strengths:

(1) The use of multiple computational models (computational fluid dynamics, CFD, for full Navier-Stokes equations and computationally-efficient inviscid vortex sheet, VS, model) offers an extra degree of reliability of the observed findings and backing to the use of simplified models for future research in more complex settings.

(2) The systematic assessment of the stability and energy savings in multiple configurations of pairs and larger ensembles of flapping foils is an important addition to the literature.

(3) The discovery of a linear phase-distance relationship in the formation attained by pairs of flapping foils is a significant contribution, which helps compare different experimental observations in the literature.

(4) The observation of a critical size effect for in-line formations, above which cohesion and energetic benefits are lost at once, is a new discovery to the field.

Weaknesses:

(1) The extent to which observations on one-degree-of-freedom flapping foils could translate to real fish schools is presently unclear, so that some of the conclusions on live fish schools are likely to be overstated and would benefit from some more biological framing.

(2) The analysis of non-reciprocal coupling is not as novel as the rest of the study and potentially not as convincing due to the chosen linear metric of interaction (that is, the flow agreement).

Overall, this is a rigorous effort on a critical topic: findings of the research can offer important insight into the hydrodynamics of fish schooling, stimulating interdisciplinary research at the interface of computational fluid mechanics and biology.

---

## [Referee Report · Reviewer #2 (Public review)]

This work makes substantial progress towards understanding physical aspects of formation locomotion, notably the hydrodynamic stability of groups of flappers and the modifications to energy costs associated with flow interactions.

Major strengths pertain to the fact that this topic is timely, interesting and complex, and the authors have advanced the understanding through their characterizations.

The weaknesses may relate to the many idealizations employed in the simulations and models, which may raise questions about how to interpret their results and whether the outcomes hold generally. But given the complexity of the problem, simplifications are necessary. The authors have certainly provided a clear presentation with appropriate details and caveats that will help the reader extract the main messages and form their own conclusions.

Overall, the work is a positive addition to the growing set of studies into schooling, flocking and related problems where unsteady flow interactions lead to interesting collective effects.

---

## [Author Response]

The following is the authors’ response to the original reviews.

**Public Reviews:**

**Reviewer #1 (Public Review):**
Summary:The study seeks to establish accurate computational models to explore the role of hydrodynamic interactions on energy savings and spatial patterns in fish schools. Specifically, the authors consider a system of (one degree-of-freedom) flapping airfoils that passively position themselves with respect to the streamwise direction, while oscillating at the same frequency and amplitude, with a given phase lag and at a constant cross-stream distance. By parametrically varying the phase lag and the cross-stream distance, they systematically explore the stability and energy costs of emergent configurations. Computational findings are leveraged to distill insights into universal relationships and clarify the role of the wake of the leading foil.

We would like to thank the referee for their careful read of the manuscript and for their constructive feedback. We appreciate it.

Strengths:(1) The use of multiple computational models (computational fluid dynamics, CFD, for full Navier-Stokes equations and computationally efficient inviscid vortex sheet, VS, model) offers an extra degree of reliability of the observed findings and backing to the use of simplified models for future research in more complex settings.(2) The systematic assessment of the stability and energy savings in multiple configurations of pairs and larger ensembles of flapping foils is an important addition to the literature.(3) The discovery of a linear phase-distance relationship in the formation attained by pairs of flapping foils is a significant contribution, which helps compare different experimental observations in the literature.(4) The observation of a critical size effect for in-line formations of larger, above which cohesion and energetic benefits are lost at once, is a new discovery in the field.

Thank you for this list of strength – we are delighted that these ideas were clearly communicated in our manuscript.

Note that Newbolt *et al*. PNAS, 2019 reported distance as a function of phase for pairs of flapping hydrofoils, and Li *et al*, Nat. Comm., 2020 also reported phase-distance relationship in robotic and biological fish (calling it Vortex Phase Matching). We compiled their results, together with our and other numerical and experimental results, showing that the linear distance-phase relationship is universal.

Weaknesses:

(1) The extent to which observations on one-degree-of-freedom flapping foils could translate to real fish schools is presently unclear so some of the conclusions on live fish schools are likely to be overstated and would benefit from some more biological framing.

Thank you for bringing up this point. Indeed, flapping foils that are free to translate in both the x- and y-directions and rotate in the x-y plane could drift apart in the y-direction. However, this drift occurs at a longer time scale than the forward swimming motion; it is much slower. For this reason, we feel justified to ignore it for the purpose of this study, especially that the pairwise equilibria in the swimming x-direction are reached at a faster time scale.

Below, we include two snapshots taken from published work from the group of Petros Koumoutsakos (Gazzola *et al*, SIAM 2014). The figures show, respectively, a pair and a group of five undulating swimmers, free to move and rotate in the x-y plane. The evolution of the two and five swimmers is computed in the absence of any control. The lateral drift is clearly sub-dominant to the forward motion. Similar results were reported in Verma *et al*, PNAS 2018.

These results are independent on the details of the flow interactions model. For example, similar lateral drift is observed using the dipole model dipole model (Kanso & Tsang, FDR 2014, Tsang & Kanso, JNLS 2023).

Another reason why we feel justified to ignore these additional degrees of freedom is the following: we assume a live fish or robotic vehicle would have feedback control mechanisms that correct for such drift. Given that it is a slowly-growing drift, we hypothesize that the organism or robot would have sufficient time to respond and correct its course.

Indeed, in Zhu *et al*. 2022, an RL controller, which drives an individual fish-like swimmer to swim at a given speed and direction, when applied to pairs of swimmers, resulted in the pair "passively" forming a stable school without any additional information about each other.

We edited the main manuscript in page 4 of the manuscript to include reference to the work cited here and to explain the reasons for ignoring the lateral drift.

Citations:

Gazzola, M., Hejazialhosseini, B., & Koumoutsakos, P. (2014). Reinforcement learning and wavelet adapted vortex methods for simulations of self-propelled swimmers. *SIAM Journal on Scientific Computing*, *36*(3), B622-B639. DOI: https://doi.org/10.1137/130943078

Verma, S., Novati, G., & Koumoutsakos, P. (2018). Efficient collective swimming by harnessing vortices through deep reinforcement learning. *Proceedings of the National Academy of Sciences*, *115*(23), 5849-5854. DOI: https://doi.org/10.1073/pnas.1800923115

Tsang, A. C. H. & Kanso, E., (2013). Dipole Interactions in Doubly Periodic Domains. *Journal of Nonlinear Science* 23 (2013): 971-991. DOI: https://doi.org/10.1007/s00332-013-9174-5

Kanso, E., & Tsang, A. C. H. (2014). Dipole models of self-propelled bodies. *Fluid Dynamics Research*, *46*(6), 061407. DOI: https://doi.org/10.1088/0169-5983/46/6/061407

Zhu, Y., Pang, J. H., & Tian, F. B. (2022). Stable schooling formations emerge from the combined effect of the active control and passive self-organization. *Fluids*, *7*(1), 41. DOI: https://doi.org/10.3390/fluids7010041

**Author response image 1. sa3fig1:** Antiphase self-propelled anguilliform swimmers. (a) – (d) Wavelet adapted vorticity fields at, respectively, t = T, t = 4T, t = 10T. (e) Absolute normalized velocities |U|/L. (f) Swimmers’ centre of mass trajectories.

**Author response image 2. sa3fig2:** Parallel schooling formation. (a) – (d) wavelet adapted vorticity fields at, respectively, t = T, t = 4T, t = 7T, t = 10T. (e) Absolute normalized velocities |U|/L. (f) Swimmers’ center of mass trajectories.

(2) The analysis of non-reciprocal coupling is not as novel as the rest of the study and potentially not as convincing due to the chosen linear metric of interaction (that is, the flow agreement).

We thank the referee for this candid and constructive feedback. In fact, we view this aspect of the study as most “revolutionary” because it provides a novel approach to pre-computing the locations of stable equilibria even without doing expensive all-to-all coupled simulations or experiments.

Basically, the idea is the following: you give me a flow field, it doesn’t matter how you obtained it, whether from simulations or experimentally, and I can tell you at what locations in this flow field a virtual flapping swimmer would be stable and save hydrodynamic energy!

In the revised version, we changed page 3 and 7 in main text, and added a new section “Diagnostic tools” in SI to better illustrate this.

Overall, this is a rigorous effort on a critical topic: findings of the research can offer important insight into the hydrodynamics of fish schooling, stimulating interdisciplinary research at the interface of computational fluid mechanics and biology.

We thank the referee again for their careful read of the manuscript and their constructive feedback.

**Reviewer #2 (Public Review):**
The document "Mapping spatial patterns to energetic benefits in groups of flow-coupled swimmers" by Heydari et al. uses several types of simulations and models to address aspects of stability of position and power consumption in few-body groups of pitching foils. I think the work has the potential to be a valuable and timely contribution to an important subject area. The supporting evidence is largely quite convincing, though some details could raise questions, and there is room for improvement in the presentation. My recommendations are focused on clarifying the presentation and perhaps spurring the authors to assess additional aspects:

We would like to thank the referee for their careful read of the manuscript and for their constructive feedback. We appreciate it.

(1) Why do the authors choose to set the swimmers free only in the propulsion direction? I can understand constraining all the positions/orientations for investigating the resulting forces and power, and I can also understand the value of allowing the bodies to be fully free in x, y, and their orientation angle to see if possible configurations spontaneously emerge from the flow interactions. But why constrain some degrees of freedom and not others? What's the motivation, and what's the relevance to animals, which are fully free?

We would like to thank the referee for raising this point. It is similar to the point raised above by the first referee. As explained above the reason is the following: in freely-swimming, hydrodynamically-interacting “fish,” the lateral drift is sub-dominant to the forward swimming motion. Therefore, we ignore it in the model. Please see our detailed response above for further clarification, and see changes in page 4 in the main manuscript.

(2) The model description in Eq. (1) and the surrounding text is confusing. Aren't the authors computing forces via CFD or the VS method and then simply driving the propulsive dynamics according to the net horizontal force? It seems then irrelevant to decompose things into thrust and drag, and it seems irrelevant to claim that the thrust comes from pressure and the drag from viscous effects. The latter claim may in fact be incorrect since the body has a shape and the normal and tangential components of the surface stress along the body may be complex.

Thank you for pointing this out! It is indeed confusing.

In the CFD simulations, we are computing the net force in the swimming x-direction direction by integrating using the definition of force density in relation to the stress tensor. There is no ambiguity here.

In the VS simulations, however, we are computing the net force in the swimming x-direction by integrating the pressure jump across a plate of zero thickness. There is no viscous drag. Viscous drag is added by hand, so-to-speak. This method for adding viscous drag in the context of the VS model is not new, it has been used before in the literature as explained in the SI section “Vortex sheet (VS) model” (pages 30 and 31).

.

(3) The parameter taudiss in the VS simulations takes on unusual values such as 2.45T, making it seem like this value is somehow very special, and perhaps 2.44 or 2.46 would lead to significantly different results. If the value is special, the authors should discuss and assess it. Otherwise, I recommend picking a round value, like 2 or 3, which would avoid distraction.

Response: The choice of dissipation time is both to model viscous effect and reduce computational complexity. Introducing it is indeed introduces forcing to the simulation. Round value, like 2 or 3, is equal to an integer multiple of the flapping period, which is normalized to T = 1, Therefore, an integer value of τdiss  would cause forcing at the resonant frequency and lead to computational blow up. To avoid this effect, a parameter choice of τdiss  = 2.45, 2.44 or 2.46 would be fine and would lead to small perturbation to the overall simulation, compared to no dissipation at all. This effect is studied in detail in the following published work from our group:

Huang, Y., Ristroph, L., Luhar, M., & Kanso, E. (2018). Bistability in the rotational motion of rigid and flexible flyers. *Journal of Fluid Mechanics*, *849*, 1043-1067. DOI: https://doi.org/10.1017/jfm.2018.446

(4) Some of the COT plots/information were difficult to interpret because the correspondence of beneficial with the mathematical sign was changing. For example, DeltaCOT as introduced on p. 5 is such that negative indicates bad energetics as compared to a solo swimmer. But elsewhere, lower or more negative COT is good in terms of savings. Given the many plots, large amounts of data, and many quantities being assessed, the paper needs a highly uniform presentation to aid the reader.

Thank you for pointing this out! We updated Figures 3,6 as suggested.

(5) I didn't understand the value of the "flow agreement parameter," and I didn't understand the authors' interpretation of its significance. Firstly, it would help if this and all other quantities were given explicit definitions as complete equations (including normalization). As I understand it, the quantity indicates the match of the flow velocity at some location with the flapping velocity of a "ghost swimmer" at that location. This does not seem to be exactly relevant to the equilibrium locations. In particular, if the match were perfect, then the swimmer would generate no relative flow and thus no thrust, meaning such a location could not be an equilibrium. So, some degree of mismatch seems necessary. I believe such a mismatch is indeed present, but the plots such as those in Figure 4 may disguise the effect. The color bar is saturated to the point of essentially being three tones (blue, white, red), so we cannot see that the observed equilibria are likely between the max and min values of this parameter.

Thank you for pointing this out! You are correct in your understanding of the flow agreement parameter, but not in your interpretation.

Basically, “if the match were perfect, then the swimmer would generate no relative flow and thus no thrust,” means that “such a location could not be is an equilibrium.” Let me elaborate. An equilibrium is one at which the net thrust force is zero. The equilibrium is stable if the slope of the thrust force is negative. Ideally, this is what maximizing the flow agreement parameter would produce.

For example, consider an ideal fluid where the flow velocity is form u(t)=2πfAucos⁡(2πft) in vertical direction. Consider a “ghost swimmer” heaving at a velocity u(t)=2πfAucos⁡(2πft) . Under this scenario, flow agreement and thrust parameters areV=AuAvcos⁡ϕ,T=−Au2−Av2+2AvAucos⁡ϕ

Let’s now consider a balance of forces on the “ghost swimmer.” The ghost swimmer is in relative equilibrium if and only if:D+T=0,cos⁡ϕ=Au2+Av2−D2AvAuϕ=±cos−1⁡Au2+Av2−D2AvAu

It gives usV=Au2+Av2−D2Av2

We then consider stability at this equilibrium by calculating the derivative of thrust parameter over phase∂T∂ϕ=−2AvAusin⁡ϕ

The corresponding values at equilibria are∂T∂ϕ=∓2AvAu1−Au2+Av2−D22AvAu

Thus, when taking the positive ϕ,∂T∂ϕ<0 which means the equilibria is a stable fixed point. We included this analysis in a new section in the SI page 32.

(6) More generally, and related to the above, I am favorable towards the authors' attempts to find approximate flow metrics that could be used to predict the equilibrium positions and their stability, but I think the reasoning needs to be more solid. It seems the authors are seeking a parameter that can indicate equilibrium and another that can indicate stability. Can they clearly lay out the motivation behind any proposed metrics, and clearly present complete equations for their definitions? Further, is there a related power metric that can be appropriately defined and which proves to be useful?

Thank you – these are excellent suggestions. Indeed, we needed to better explain the motivation and equations. Perhaps the main idea for these metrics can be best understood when explained in the context of the simpler particle model, which we now do in the SI and explain the main text.

(7) Why do the authors not carry out CFD simulations on the larger groups? Some explanations should be given, or some corresponding CFD simulations should be carried out. It would be interesting if CFD simulations were done and included, especially for the in-line case of many swimmers. This is because the results seem to be quite nuanced and dependent on many-body effects beyond nearest-neighbor interactions. It would certainly be comforting to see something similar happen in CFD.

We are using a open-source version of the Immersed Boundary Method that is not specifically optimized for many interacting swimmers. Therefore, the computational cost of performing CFD simulations for more swimmers is high. Therefore, we used the CFD simulations sporadically with fewer simmers (2 or 3) and we performed systematic simulations in the context of the VS model.

For the same Reynolds number in Figure 1, we simulated three and four swimmers in CFD: three swimmers forms a stable formation, four swimmers don’t, consistent with the VS model, with the forth swimmer colliding with the third one. Results are included in the SI figure 8 of the main text.

(8) Related to the above, the authors should discuss seemingly significant differences in their results for long in-line formations as compared to the CFD work of Peng et al. [48]. That work showed apparently stable groups for numbers of swimmers quite larger than that studied here. Why such a qualitatively different result, and how should we interpret these differences regarding the more general issue of the stability of tandem groups?

Thank you for bringing up this important comparison. Peng et al. [48] (Hydrodynamic schooling of multiple self-propelled flapping plates) studied inline configuration of flapping airfoils at Reynolds number = 200. There are several differences between their work and ours. The most important one is that they used a flexible plate, which makes the swimmer more adaptive to changes in the flow field, e.g. changes in tailbeat amplitude and changes in phase along its body and diverts some of the hydrodynamic energy to elastic energy. We edited the main text page 10 at the end of section “Critical size of inline formations beyond which cohesion is lost” to explain this distinction.

(9) The authors seem to have all the tools needed to address the general question about how dynamically stable configurations relate to those that are energetically optimal. Are stable solutions optimal, or not? This would seem to have very important implications for animal groups, and the work addresses closely related topics but seems to miss the opportunity to give a definitive answer to this big question.

Indeed, that is exactly the point – in pairwise formations, stable configurations are also energetically optimal! In larger groups, there is no unique stable configuration – each stable configuration is associated with a different degree of energy savings. Interestingly, when exploring various equilibrium configurations in a school of four, we found the diamond formation of D. Weihs, Nature, 1972 to be both stable and most optimal among the configurations we tested. However, claiming this as a global optimum may be misleading – our standpoint is that fish schools are always dynamic and that there are opportunities for energy savings in more than one stable configuration.

We added a section in new text “Mapping emergent spatial patterns to energetic benefits”, and added a new figure in the maintext (Fig. 10) and a new figure in the SI (Fig. S. 8)

(10) Time-delay particle model: This model seems to construct a simplified wake flow. But does the constructed flow satisfy basic properties that we demand of any flow, such as being divergence-free? If not, then the formulation may be troublesome.

The simplified wake flow captures the hydrodynamic trail left by the swimmer in a very simplified manner. In the limit of small amplitude, it should be consistent with the inviscid vortex sheet shed of T. Wu’s waving swimmer model (Wu TY. 1961).

The model was compared to experiments and used in several recent publications from the Courant Institute (Newbolt *et al*. 2019, 2022, 2024).

Citations:

Wu, T. Y. T. (1961). Swimming of a waving plate. *Journal of Fluid Mechanics*, *10*(3), 321-344. DOI: https://doi.org/10.1017/S0022112061000949

Newbolt, J. W., Lewis, N., Bleu, M., Wu, J., Mavroyiakoumou, C., Ramananarivo, S., & Ristroph, L. (2024). Flow interactions lead to self-organized flight formations disrupted by self-amplifying waves. *Nature Communications*, *15*(1), 3462. DOI: https://doi.org/10.1038/s41467-024-47525-9

Newbolt, J. W., Zhang, J., & Ristroph, L. (2022). Lateral flow interactions enhance speed and stabilize formations of flapping swimmers. *Physical Review Fluids*, *7*(6), L061101. DOI: https://doi.org/10.1103/PhysRevFluids.7.L061101

Newbolt, J. W., Zhang, J., & Ristroph, L. (2019). Flow interactions between uncoordinated flapping swimmers give rise to group cohesion. *Proceedings of the National Academy of Sciences*, *116*(7), 2419-2424. DOI: https://doi.org/10.1073/pnas.1816098116

**Recommendations for the authors:**

**Reviewer #1 (Recommendations For The Authors):**
Congratulations on such a comprehensive and well-thought-out study; I truly enjoyed reading it and have only a couple of suggestions that I believe will help further strengthen the paper. I am including a bunch of references here that are very familiar to me without the expectation of you to include them all, just to point at areas that I feel you might consider useful.

We thank the referee again for their careful read of the manuscript and for their constructive feedback. We appreciate it.

First, I believe that some more rationale is needed to justify the chosen modeling framework. I am fully aware of how difficult is to run these simulations, but I see some critical assumptions that need to be at least spelled out for the reader to appreciate the limitations of the study: (1) Constraining the cross-stream coordinate (a stability analysis should include perturbations on the cross-stream coordinate as well, see, for example, https://doi.org/10.1017/flo.2023.25 -- I know this is much simpler as it discards any vortex shedding) and (2) Assuming equal frequency and amplitude (there are studies showing variation of tail beat frequency in animals depending on their position in the school, see, for example, https://doi.org/10.1007/s00265-014-1834-4).

Thank you for these suggestions. These are indeed important and interesting points to discuss in the manuscript. See response above regarding point 1. Regarding point 2, this is of course important and will be pursued in future extensions of this work. We edited the intro and discussion of the main text to explain this.

In the paper “Stability of schooling patterns of a fish pair swimming against a flow”, The authors considered a pair of swimmers swimming in a channel. They analyzed stability of the system and find multiple equilibria of the system, including inline and staggered formation, and a special formation of perpendicular to the wall. Studying fish school in confined domain and analyzing their stability is very interesting. We added citation to this paper in the discussion section at the end of page 10.

In the paper “Fish swimming in schools save energy regardless of their spatial position”, the authors measured the reduction in power of fish by measuring tail beat frequency and oxygen consumption and compared them to measurements in solitary fish. They found that in a school of fish, individuals always save power comparing to swimming alone. However, there is one important caveat in this study: they considered a larger school of fish and expressed the results in terms of pairwise configurations (see schematics we draw below). This is misleading because it may suggest that formations with only two fish provide benefits each other, while in fact, the data is obtained from a larger school with many neighbors. They only consider a fish’s relationship to its nearest neighbor. But in a large school, other neighbors will also have influence on their energy consumption. In the schematics below, we emphasized on several focal fishes, marking them as red, green, and blue. We also marked their nearest neighbors using the same color, but lighter. The nearest neighbors are what the authors are considering to show its neighbor relationship. For example, a problematic one is the red fish, for which its nearest neighbor is behind it, but indeed, its power saving may come from the other neighbors, which are around or ahead it.

**Author response image 3. sa3fig3:** 

Second, I would like to see more biology context with respect to limitations that are inherent to a purely mechanical model, including, neglecting vision that we know plays a synergistic role in determining schooling patterns. For example, a recent study https://doi.org/10.1016/j.beproc.2022.104767 has presented experiments on fish swimming in the dark and in bright conditions, showing that it is unlikely that hydrodynamics alone could explain typically observed swimming patterns in the literature.

Thank you for this suggestion and for sharing us with the paper “Collective response of fish to combined manipulations of illumination and flow”. This is a great study, and we are sorry to have missed it.

In this paper, the authors found that when having illumination, fish swim more cohesively, which is in consistent with another paper we already cited “The sensory basis of schooling by intermittent swimming in the rummy-nose tetra (Hemigrammus rhodostomus)”. Another important conclusion in this paper is that when having brighter illumination and with flow, fish school spend more time side by side. This connects well to the conclusion in another paper we cited “Simple phalanx pattern leads to energy saving in cohesive fish schooling,” where at lower flow speed in a water channel, fish tended to form a dynamic school while at higher flow speed, they organized in a side-by-side/ phalanx configuration. This conclusion is consistent with our study that in side-by-side formation, fish share power saving.

Importantly, it is well known that both vision and flow sensing play important roles in fish schooling. This study aimed to merely explore what is possible through passive hydrodynamic interactions, without visual and flow sensing and response. We clarify this in the revised version of the manuscript.

Third, I am not too convinced about the flow agreement metric, which only accounts for linear interactions between the foils. More sophisticated approaches could be utilized as the one proposed here https://doi.org/101.1017/jfm.2018.369, based on a truly model-agnostic view of the interaction - therein, the authors show non-reciprocal (in strength and time-scale) coupling between two in-line flapping foils using information theory. I also would like to mention this older paper https://doi.org/10.1098/rsif.2012.0084, where an equivalent argument about the positioning of a trailing fish with respect to a leading robotic fish is made from experimental observations.

Thank you for these remarks and for sharing these two interesting papers.

The flow agreement metric is not specific to two fish, as we show in Fig. 6 of the manuscript. We edited the manuscript and SI to better explain the motivation and implementation of the flow agreement parameter. We edited the main text, see revisions on page 7, and added a new section call “diagnostic tools.”.

In the paper “An information-theoretic approach to study fluid–structure interactions”, the authors calculate the transfer entropy between two oscillating airfoils when they are hydrodynamically coupled. This is an interesting study! We will apply this approach to analyzing larger schools in the future. We cited this paper in the introduction.

In the paper “Fish and robots swimming together: attraction towards the robot demands biomimetic locomotion”, the authors found that fish will swim behind an artificial fish robot, especially when the fish robot is beating its tail instead of static. At specific conditions, the fish hold station behind the robot, which may be due to the hydrodynamic advantage obtained by swimming in the robot’s wake. DPIV resolved the wake behind a static/ beating fish robot, but did not visualize the flow field when the fish is there. This study is similar to a paper we already cited “In-line swimming dynamics revealed by fish interacting with a robotic mechanism”, in which, they considered fish-foil interaction. In the revised manuscript, we cite both papers.

For the reviewer’s comments about flow agreement only accounts for linear interactions between the foils, we want to explain more to clarify this. The flow agreement parameter is a nonlinear metric, which considered the interaction between a virtual swimmer and an arbitrary unsteady flow field. Although the metric is a linear function of swimmer’s speed, it is indeed a nonlinear function of spacing and phase, which are the quantities we care about. Moreover, the flow field can by generated by either experiment or CFD simulation, and behind one or more swimmers. It is true that it is a one way coupled system since the virtual swimmer does not perturb the flow field.

Again, this is great work and I hope these suggestions are of help.

Thank you again! We are delighted to receive such a positive and constructive feedback.

**Reviewer #2 (Recommendations For The Authors):**
(1) About Figure 1: Panel C should be made to match between CFD and VS with regard to the swimmer positions. Also, if the general goal of the figure is to compare CFD and VS, then how about showing a difference map of the velocity fields as a third column of panels across A-D?

Thank you for pointing this out. Figure 1 C is updated accordingly.

The general goal is to show the CFD and VS simulations produce qualitatively similar results. Some quantities are not the same across models, e.g. the swimming speed of swimmers are different, but the scaled distance is the same.

(2) Figure 3: In A, it would be nice to keep the y-axis the same across all plots, which would aid quick visual comparison. In B, the legend labels for CFD and VS should be filled in with color so that the reader can more easily connect to the markers in the plot.

Thank you for pointing this out, we’ve updated figure 3 and 6.

(3) Figures 4, 9, and Supplementary Figures too: As mentioned previously, the agreement parameter plots are saturated in the color map, possibly obscuring more detailed information.

Thank you for pointing this out. The goal is to show that there is a large region with positive flow agreement parameter.

We picked up the flow agreement behind a single swimmer in VS simulation (Fig.4B) and added the counter lines to it (represents 0.25 and 0.5). Not many details are hidden by the saturated colormap.

**Author response image 4. sa3fig4:** 

We also updated Fig 4 and Fig 9 accordingly.

(4) Figure 6: Is this CFD or VS? Why show one or the other and not both? In B, it seems that there are only savings available and no energetically costly positions. This seems odd. In C, it seems the absolute value on dF/dd is suppressing some important information about stability - the sign of this seems important. In E, the color bar seems to be reflected from what is standard, i.e. 0 on the left and 100 on the right, as in F.

Thank you for asking. Fig. 6 is based only on VS simulations. There are hundreds of simulations in this figure, we are not running CFD simulations to save computational effort. Representative CFD simulations are shown in Figure 1,2,3, for comparison. We added a sentence in the figure caption for clarification.

In C, since ∂F∂d is always negative for emergent formations (only stable equilibria can appear during forward time simulation), we are showing its absolute value for comparison.

In E, we are flipping this because larger flow agreement parameter corresponds to more power saving, in the other word, negative changes in COT.

(5) Fig. 8: For cases such as in D that have >100% power savings, does this mean that the swimmer has work done by the flow? How to interpret this physically for a flapping foil and biologically for a fish?

Yes, it means the hydrofoil/fish gets a free ride, and even able to harvest energy from the incoming flow. Actually, similar phenomenon has been reported in the biology and engineering literature. For example, Liao *et al*. 2003, Beal *et al*. 2006 found that live or dead fish can harvest energy from incoming vortical flow by modulating their body curvature.

In engineering, Chen *et al*. 2018, Ribeiro *et al*. 2021 have found that the following airfoil in a tandem/ inline formation can harvest energy from the wake of leading swimmer in both simulation and experiemnts.

Citations:

Liao, J. C., Beal, D. N., Lauder, G. V., & Triantafyllou, M. S. (2003). Fish exploiting vortices decrease muscle activity. *Science*, *302*(5650), 1566-1569. DOI: https://doi.org/10.1126/science.1088295

Beal, D. N., Hover, F. S., Triantafyllou, M. S., Liao, J. C., & Lauder, G. V. (2006). Passive propulsion in vortex wakes. *Journal of fluid mechanics*, *549*, 385-402. DOI: https://doi.org/10.1017/S0022112005007925

Chen, Y., Nan, J., & Wu, J. (2018). Wake effect on a semi-active flapping foil based energy harvester by a rotating foil. *Computers & Fluids*, *160*, 51-63. DOI: https://doi.org/10.1016/j.compfluid.2017.10.024

Ribeiro, B. L. R., Su, Y., Guillaumin, Q., Breuer, K. S., & Franck, J. A. (2021). Wake-foil interactions and energy harvesting efficiency in tandem oscillating foils. *Physical Review Fluids*, *6*(7), 074703. DOI: https://doi.org/10.1103/PhysRevFluids.6.074703